# Transcriptome Analysis and Identification of Lipid Genes in *Physaria lindheimeri*, a Genetic Resource for Hydroxy Fatty Acids in Seed Oil

**DOI:** 10.3390/ijms22020514

**Published:** 2021-01-06

**Authors:** Grace Q. Chen, Won Nyeong Kim, Kumiko Johnson, Mid-Eum Park, Kyeong-Ryeol Lee, Hyun Uk Kim

**Affiliations:** 1Western Regional Research Center, Agricultural Research Service, U.S. Department of Agriculture, Albany, CA 94710, USA; kumiko.johnson@usda.gov; 2Department of Bioindustry and Bioresource Engineering, Sejong University, Seoul 05006, Korea; rladnjssud@naver.com; 3Department of Molecular Biology, Graduate School, Sejong University, Seoul 05006, Korea; aledja@naver.com; 4Department of Agricultural Biotechnology, National Institute of Agricultural Sciences, Rural Development Administration, Jeonju 54974, Korea; realdanny@korea.kr

**Keywords:** hydroxy fatty acid, *Physaria lindheimeri*, seed, transcriptome, triacylglycerol, gene expression

## Abstract

Hydroxy fatty acids (HFAs) have numerous industrial applications but are absent in most vegetable oils. *Physaria lindheimeri* accumulating 85% HFA in its seed oil makes it a valuable resource for engineering oilseed crops for HFA production. To discover lipid genes involved in HFA synthesis in *P. lindheimeri*, transcripts from developing seeds at various stages, as well as leaf and flower buds, were sequenced. Ninety-seven percent clean reads from 552,614,582 raw reads were assembled to 129,633 contigs (or transcripts) which represented 85,948 unique genes. Gene Ontology analysis indicated that 60% of the contigs matched proteins involved in biological process, cellular component or molecular function, while the remaining matched unknown proteins. We identified 42 *P. lindheimeri* genes involved in fatty acid and seed oil biosynthesis, and 39 of them shared 78–100% nucleotide identity with Arabidopsis orthologs. We manually annotated 16 key genes and 14 of them contained full-length protein sequences, indicating high coverage of clean reads to the assembled contigs. A detailed profiling of the 16 genes revealed various spatial and temporal expression patterns. The further comparison of their protein sequences uncovered amino acids conserved among HFA-producing species, but these varied among non-HFA-producing species. Our findings provide essential information for basic and applied research on HFA biosynthesis.

## 1. Introduction

Hydroxy fatty acids (HFAs) and their derivatives are used as raw materials for numerous industrial products, such as lubricants, plastics and surfactants [1,2]. The conventional source of HFA is from castor (*Ricinus communis*), which contains 90% ricinoleic acid (12-hydroxy 9-octadecenoic acid, 18:1OH) in its seed oil. The production of castor oil, however, is hampered by the presence of the toxin ricin [3,4] and hyper-allergic 2S albumins [5,6,7] in its seeds. Lesquerella (*Physaria fendleri*) and *Physaria lindheimeri* seed oils contain lesquerolic acid (14-hydroxy 11-eicosenoic acid, 20:1OH), at approximately 60% [8,9,10,11,12] and 85% [13,14,15], respectively. Lesquerella possesses key domestication traits such as indehiscent seeds (or non-shattering seeds), photoperiod neutrality, upright growth habit, and extensive branching that leads to high seed yields, thus it is being developed as a safe source for HFA production [10]. Considerable breeding efforts, such as single plant or bulking recurrent selections, are made to improve lesquerella with increased HFA oil content [12]. *P. lindheimeri* is also crossed with lesquerella, but there is no significant increase in HFA level in the hybrid off-springs of lesquerella [16]. With the success of lesquerella biotechnology [17,18], genes and regulatory elements from *P. lindheimeri* can be used as excellent targets for improving lesquerella through *Agrobacterium*-mediated genetic transformation. Besides the HFAs found in plant seeds, families of saturated hydroxy fatty acids (SHFAs) have been recently discovered in cow, goat, and human milks [19,20], and the SHFAs from human milk inhibit the growth of human cancer cells and suppress beta-cell apoptosis [19], indicating that SHFAs may play a role in the promotion and protection of human health.

The formation of seed oil (triacylglycerol, TAG) starts from *de novo* fatty acid (FA) biosynthesis in plastid and TAG assembly in the endoplasmic reticulum (ER) [21]. HFA formation and HFA-containing TAG assembly occur after the FA is exported to cytosol and activated to acyl-coenzyme A (acyl-CoA) [22]. Based on the current knowledge, a proposed pathway of 20:1OH synthesis and 20:1OH-TAG assembly in *P. lindheimeri* is depicted in Figure 1. In developing *P. lindheimeri* seeds, oleic acid (18:1) is synthesized in the plastid, exported and activated to 18:1-CoA in cytosol. The newly synthesized 18:1-CoA is acylated directly into membrane lipid phosphatidylcholine (PC) in the ER by the forward reaction of lyso-PC acyltransferase (LPCAT) [23,24,25], resulting in 18:1-PC (purple arrow in Figure 1). As 18:1-PC is the substrate of oleate 12-hydroxylase (FAH12) [26,27,28], a *P. lindheimeri* FAH12 (PlFAH12) [13] hydroxylates 18:1-PC to form 18:1OH-PC (Figure 1). Through the reverse reaction of LPCAT, the 18:1OH can be removed from PC and transferred back to cytosol to be activated as 18:1OH-CoA (purple arrow in Figure 1). The de-acylation of *sn*-2 PC can also occur with a phospholipase A (PLA_2_)-type activity to yield a free FA, which is then activated to acyl-CoA [29] (orange arrow in Figure 1). In *Physaria* species accumulating 20:1OH, an FA-condensing enzyme KCS18 (or KCS3) elongates 18:1OH-CoA to 20:1OH-CoA [30] (Figure 1). Rapid acylation and de-acylation (or acyl editing) by LPCAT (or by PLA_2_), and in conjunction with efficient elongation by KCS18, lead to the enrichment of 20:1OH-CoA in the cytosol during seed development. Once the 20:1OH-CoA is formed, 20-1OH-CoA can be acylated to TAG by the conventional Kennedy pathway [31,32,33], which consists of three sequential acylations of acyl-CoAs to a glycerol-3-phosphate (G3P) backbone (solid blue arrows in Figure 1). First, the *sn*-1 position of G3P is acylated by glycerol-3-phosphate acyltransferase (GPAT) to produce lysophosphatidic acid (LPA). Second, the *sn*-2 position of LPA is acylated by LPA acyltransferase (LPAT) to generate phosphatidic acid (PA). PA is then converted to 1,2-*sn*-diacylglycerol (DAG, or de novo DAG) by PA phosphatase (PAP). Finally, the *sn*-3 position of DAG is acylated by 1,2-*sn*-diacylglycerol acyltransferase (DGAT) to produce TAG. In addition to the Kennedy pathway, the 20:1OH-CoA can also be directly acylated to PC by LPCAT, generating 20:1OH-PC (purple arrow in Figure 1), and the 20:1OH on PC can be transferred to the *sn*-3 position of DAG by phospholipid:DAG acyltransferase (PDAT), forming 20:1OH-TAG [34,35,36] (solid brown arrows in Figure 1).

Metabolic labeling experiments reveal that in some oilseeds containing high amounts of poly-unsaturated FA, such as linoleic acid (18:2) and linolenic acid (18:3), a separate PC-derived DAG pool is utilized for TAG synthesis [25]. Like 18:1OH, 18:2 and 18:3 are also formed on PC through the desaturation of 18:1-PC to 18:2-PC, and 18:2-PC to 18:3-PC by FA desaturase 2 (FAD2) [37] and FA desaturase 3 (FAD3) [38,39], respectively. These modified FA-PCs (including 18:2-PC, 18:3-PC, and 18:1OH-PC) can be converted to PC-derived DAG mainly by PC:DAG cholinephosphotransferase (PDCT) through the phosphocholine headgroup exchange between PC and DAG [25,40,41] (green arrow in Figure 1). Alternatively, PC-derived DAG can be produced by the reverse action of CDP-choline:DAG cholinephosphotransferase (CPT) [42], or a lipase-based mechanism using phospholipase C, or phospholipase D plus PAP [25]. By this means, PC-derived DAG can be enriched with modified FA, such as 18:2-, 18:3-, and 18:1OH-DAG. Thus, utilizing PC-derived DAG represents a mechanism of channeling modified FA to TAG [25]. In case of *P. lindheimeri*, the modified 18:1OH has to be removed from PC and elongated to 20:1OH in cytosol, so 20:1OH-PC can be formed by the direct acylation of 20:1OH-CoA to LPC by LPCAT (purple arrows in Figure 1). The LPCAT-mediated formation of 20:1OH-PC and PC-derived 20:1OH-DAG synthesis would be regarded as new substrates for the synthesis of TAG catalyzed by PDAT or DGAT (dotted brown and blue arrows, Figure 1).

Castor genes are introduced into non-HFA-oilseed Arabidopsis (*Arabidopsis thaliana*) or Camelina (*Camelina Sativa*) to study the HFA biosynthesis mechanism [43]. Castor *RcFAH12* is firstly isolated and demonstrated to be responsible for HFAs synthesis in transgenic seeds up to 17% [28,44,45]. Additional genes, Castor *RcDGAT2* [46], *RcPDAT1-2* (or *RcPDAT1-A*) [35,36], and *RcPDCT* [41], *RcPLCL1* [47], *RcLPAT2* [48,49], *RcLPAT3B* and *RcLPATB* [49], and *RcGPAT9* together with *RcLPAT2* and *RcPDAT1A* [50], are demonstrated to increase the HFAs content of transgenic Arabidopsis or Camelina from 17% to 28%. Recent attempts at the co-expression of certain *PLA*, including *RcsPLA_2_α* [29] or *RcpPLAIIIβ* [51], with *RcFAH12* in Arabidopsis results in decreases in seed HFAs contents when compared with Arabidopsis expressing *RcFAH12* alone, which indicates that the cleaved HFAs at the *sn*-2 position of PC are not incorporated into seed TAG. However, when a castor lecithin:cholesterol acyltransferase-like PLA (*RcLCAT-PLA*) or a lesquerella *PfLCAT-PLA* is co-expressed with *RcFAH12* in Arabidopsis, the seed HFAs content remains constant, and the HFAs are re-distributed from the *sn*-2 position of TAG to the *sn*-1/3 positions of TAG [52].

Castor TAGs contain 90% 18:1OH, and over 70% of the TAGs have all three positions esterified with 18:1OH [53]. Lesquerella TAGs contain ~60% 20:1OH and almost all of it is esterified to the *sn*-1 and *sn*-3 positions [54] (Figure 1). The absence of HFAs at the *sn*-2 position of TAG might be caused by the preference of endogenous lesquerella LPATs (PfLPATs) for common FAs [18]. To increase the HFA content at the *sn*-2 position of lesquerella TAGs, we have overexpressed *RcLPAT2* in lesquerella seeds, and the resulting transgenic seed oil increases 3-HFA-TAGs (TAGs with all three *sn* positions acylated with HFAs) from 5% to 13–14% [18]. Regiochemical analysis reveals that RcLPAT2 increases the 3-HFA-TAGs content by enhancing the acylation of 18:1OH at the *sn*-2 position of 20:1OH-LPA for the subsequent generation of *sn*-1/3-20:1OH-*sn*-2-18:1OH-TAG [55]. The limited accumulation of HFAs in transgenics indicates that additional genes are required to implement new strategies for increased HFA production in lesquerella. *P. lindheimeri* oil contains 85% 20:1OH, similarly high to that of castor oil, and genetically, *P. lindheimeri* is closely related to lesquerella, both belonging to the *Physaria* genus. The genes from *P. lindheimeri* may be readily adapted to the molecular machinery of gene transcription and translation in lesquerella. Genomic data from castor [56,57] and lesquerella [58,59] are freely available; however, so far, only one gene encoding oleate 12-hydroxylase from *P. lindheimeri* (*PlFAH12*) has been isolated by polymerase chain reaction (PCR) [13]. Transgenic Arabidopsis expressing *PlFAH12* accumulated 18% HFAs in seed oil [13].

In this study, we describe *P. lindheimeri* transcriptomes constructed from leaf, flower bud, and developing seeds at various stages. The lipid genes involved in FA and TAG biosynthesis were identified by the deep mining of the transcriptomes. Detailed spatial and temporal expression profiles of key genes were characterized by quantitative PCR (qPCR). *P. lindheimeri* sequences were further compared with orthologs from HFA-producing castor and lesquerella, as well as orthologs from non-HFA-producing species, including Arabidopsis and Camelina. Our findings provide essential information for future basic and applied research on HFA biosynthesis in oilseeds.

## 2. Results and Discussion

### 2.1. Analysis of Transcriptomes, Gene Ontology and Differentially Expressed Genes (DEGs)

We have previously conducted a series of seed developmental studies in *P. lindheimeri*, including changes of weight, lipid accumulation, and FA composition [15]. We identified three major periods: (1) early period, up to 21 days after pollination (DAP), when developing seeds rapidly increase in size and fresh weight; (2) mid-maturation period, from 28 to 42 DAP, when lipids and dry weights accumulate steadily, but seed size and fresh weigh cease at 42 DAP and begin to decline; and (3) late-maturation/desiccation stages, from 42 DAP to 56 DAP, when seeds enter the natural dehydrate process. The major FA 20:1OH accumulates steadily from 28 DAP to 56 DAP, eventually reach up to 77% of the total seed oil [15].

To obtain a comprehensive profile of the transcriptome of *P. lindheimeri* seeds, we collected RNA samples from seeds at 21, 35, and 42 DAP, which represent the time points of the early, middle and late stages of seed development. RNA samples were also collected from leaf and flower buds for comparative studies. Using the Illumina Hiseq2000 system (Illumina, San Diego, CA, USA), a total of 552,614,582 raw reads were generated. After trimming the adapter and those shorter than 25 bp low-quality reads, 539,709,492 clean reads were obtained (Table 1). The number of clean reads ranged from 95,336,514 to 125,026,944, or 97% to 99% of raw reads, indicating that high quality sequencing data were equally obtained from these five samples (Table 1). The de novo reconstruction of RNA-seq data resulted in a total of 129,633 contigs (or transcripts) represented by 85,948 unique genes with an average N50 contig size (50% of all bases reside in a contig) of 1373 bp (Table 2), which is comparable to the 1310 bp of the high-quality transcriptome of *P. fendleri*, a relative of *P. lindheimeri*, which also produces 20:1OH at 55–60% in seeds [58]. Gene Ontology (GO) analysis was performed through the BlastX program at NCBI [60], which compares the six-frame conceptual translation products of a nucleotide query sequence (both strands) against a protein sequence database (go_v20150407). Among the 129,633 contigs, 40% of them did not match proteins with known functions, and the remaining 24%, 19% and 17% matched proteins in biological processes, cellular components and molecular functions, respectively (Appendix A). In the category of “biological process”, “metabolic process” (23%), “biological regulation” (14%), “response to stimulus” (12%), and “single-organism process” (8%) were major groups (Appendix A). In the category of “cellular component”, “cell part” had a largest percent of contigs (42%), followed by “organelle” (27%) (Appendix A). Within the category of “molecular function”, “binding” (39%) and “catalytic activity” (31%) counted as the two top sub-categories (Appendix A). The results of the GO analysis are similar to those for *P. fendleri* [58].

To infer the relative expression level of a gene, FPKM (fragments per kilobase of transcript per million mapped reads) was used to estimate transcript abundances [61]. After filtering out of 72,932 contigs which had an FPKM value of zero in at least one of all five samples (Materials and Methods), 56,701 remaining contigs were subjected to comparative studies. Compared with the expression level in leaf, a total of 7857 contigs showed significant changes, i.e., four-fold or more (up or down) in at least one of the developing seed or flower bud samples (Appendix A; Appendix A). Developing seeds at 21, 35 and 42 DAP had similar numbers of up-regulated contigs ranging from 2080 to 2224, and down-regulated contigs ranging from 2593 to 3082, whereas the flower bud had only 26 up- and 193 down-regulated contigs (Appendix A; Appendix A). The results suggest that the gene expression profiles in developing seeds are distinctly different from those of the leaf and flower bud. To visualize the differential expression of genes, we constructed a heat map of hierarchically clustered expression data (Figure 2). The hierarchical clustering of differentially expressed contigs clearly showed that developing seeds vs. leaf and flower bud had different expression patterns. Approximately 45% of the total contigs were up-regulated in leaf/flower buds and down-regulated in 21/42 DAP seeds (group A of clusters, Figure 2). Oppositely, about 26% of the contigs were down-regulated in leaf/flower buds and up-regulated in 21/42 DAP seeds (group D of clusters, Figure 2). The expression patterns in 35 DAP seeds were overall similar to those of 21/42 DAP seeds, with the exception of the group B of clusters which had a pattern similar to that of leaf/flower bud (Figure 2). Besides this, 35 DAP seed had other distinct patterns: about 10% of the total contigs were highly up-regulated (blue bordered contigs from group C, Figure 2), while about 5% were extremely down-regulated (yellow bordered contigs from group A, Figure 2). Most contigs in the group D from 35 DAP seed had mixed patterns, and some of them had similar levels to that of the 21/42 seeds or leaf/flower bud, and some in between (Figure 2). As 35 DAP seeds are at the peak of seed development [15], it is not a surprise to see some genes with unique differential expression patterns. For example, *PlFAH12* and *PlKCS18*’s transcript levels were highly elevated at 35 DAP, which was verified by qPCR (Figure 3 and Figure 6), suggesting that gene transcription played a critical role in regulating HFA synthesis during seed development.

### 2.2. Mining and Characterization of Genes Involved in FA and TAG Biosynthesis during P. lindheimeri Seed Development

To understand HFA biosynthesis and metabolism in *P. lindheimeri*, we searched the contigs for genes involved in FA and TAG biosynthesis. Based on known Arabidopsis genes listed in the acyl-lipid metabolism database (https://www.arabidopsis.org/browse/genefamily/acyl_lipid.jsp), we identified 42 *P. lindheimeri* orthologous contigs involved in these processes (Appendix A). Thirty-nine of these contigs shared high sequence identity with Arabidopsis genes ranging from 85 to 98%, with two genes, *KASII* and *DGAT3*, sharing 70.8% and 83.2% identity, respectively (Appendix A). The sequence similarity between *P. lindheimeri* and Arabidopsis (Appendix A) is comparable to that between *P. fendleri* and Arabidopsis [58]. The very high sequence identity among Arabidopsis *P. fendleri* and *P. lindheimeri* genes suggests a similarly high degree of conservation of their gene functions in seeds that facilitates the translation of research findings from Arabidopsis or *P. lindheimeri* to lesquerella and other oilseeds for crop improvement.

To further characterize *P. lindheimeri* genes, we examined the expression profiles of genes in various organs, including developing seeds, flower buds and leaves. The genes involved in FA biosynthesis in plastids were expressed in all these organs, indicating their house-keeping functions. Since 20:1OH and 20:1OH-TAG synthesis occurred after 18:1 being synthesized in plastid and exported to the ER for hydroxylation and elongation, we analyzed in more detail for the ER-localized genes associated with 20:1OH formation and 20:1OH-TAG assembly.

### 2.3. De novo Fatty Acid Biosynthesis in Plastids

We examined our collection of *P. lindheimeri* contigs for genes encoding the known steps of FA biosynthesis in plastids. De novo FAs are synthesized in plastids [21]. A plastidial pyruvate dehydrogenase complex (PDHC) catalyzes the conversion of pyruvate to acetyl-CoA, an initial substrate for FA synthesis (Appendix A). PDHC is a large multienzyme containing three components: E1 (pyruvate dehydrogenase or PDH, composed of E1ɑ and E1ß subunits), E2 (dihydrolipoyl acyltransferase or DHLAT), and E3 (dihydrolipoamide dehydrogenase or LPD) [21]. In *P. lindheimeri*, we identified five subunit genes for *PlPDHC*: *PDH (E1ɑ)*, *PDH (E1ß)*, *EMB3003 (E2)*, *LTA2(E2)* and *LPD1 (E3)* (Appendix A). Transcriptome analysis indicated that these five subunits were all expressed in developing seeds, leaves and flower buds (Appendix A). Once acetyl-CoA is synthesized, a heteromeric complex enzyme, acetyl-CoA carboxylase (ACC), catalyzes acetyl-CoA to form malonyl-CoA, and then a malonyl-CoA acyl-carrier protein (ACP) transferase (MCMT) further converts malonyl-CoA to malonyl-ACP (Appendix A). ACC is composed of four subunits; three of them, biotin carboxyl-carrier protein (BCCP), biotic carboxylase (BC) and alpha-carboxyltransferase (α-CT), are encoded from the nuclear genome, and beta-carboxyltransferase (β-CT) is encoded in the plastid genome [21,62]. *P lindheimeri* genes *PlBCCP1*, *PlBCCP2*, *Plα-CT* and *PlMCMT* were identified, and they exhibited a ubiquitous expression pattern in leaves, flower buds and developing seeds at various stages (Appendix A). To synthesize de novo 16- or 18-carbon FA, the initial condensation reaction of acetyl-CoA and malonyl-ACP is catalyzed by KAS isoform III (KASIII), yielding a four-carbon product (3-ketobutyrl-ACP). Subsequent condensations (up to 16:0-ACP) require KASI, whereas the final elongation of the 16-carbon palmitoyl-ACP to the 18-carbon stearoyl-ACP is catalyzed by KASII. In addition, each FA elongation cycle requires the participation of two reductases and a dehydrase. The 3-ketoacyl-ACP is first reduced by a 3-ketoacyl-ACP reductase (KAR), which is then subjected to dehydration by the enzyme hydroxyacyl-ACP dehydratase (HAD), and the enoyl-ACP thus obtained is finally reduced by the enzyme enoyl-ACP reductase (ENR) to form a saturated FA [21] (Appendix A). With the exception of *PlKASI*, we detected the expression of all the above genes in leaves, flower buds, and developing seeds from *P. lindheimeri* (Appendix A). Once FA is elongated to palmitic (16:0) or stearic acid (18:0), FA thioesterase B (FATB) cleaves acyl groups from 16:0-ACP or 18:0-ACP for the export from the plastid (Appendix A). In plastid, stearoyl-ACP desaturase (SAD) catalyzes the conversion of 18:0-ACP to 18:1-ACP, which is then cleaved by FATA (Appendix A). Arabidopsis has seven *SAD* family genes, including *FAB2* (At2g43710) and *DES6* (At4g30950) [21]. In *P. lindheimeri*, we identified *PlFATB*, *PlFATA*, *PlFAB2* and *PlDES6*, showing constitutive expression patterns in leaf, flower bud and developing seeds at various stages (Appendix A). Long-chain acyl-CoA synthase (LACS) esterifies free FAs to acyl-CoAs, a key activation step for FA to be utilized by many acyl-CoA-dependent acyltransferases. Arabidopsis plastid-localized LACS9 (At1g77590) and ER-localized LACS8 (At2g04350) have been reported [63,64]. In *P. lindheimeri* transcriptomes, we identified both *PlLACS8* and *PlLACS9* as showing spatial and temporal expression profiles ubiquitously expressed in all samples examined (Appendix A). The overall expression profiles of the genes involved in FA biosynthesis in plastids exhibited constitutive patterns, indicating their house-keeping functions in various organs.

### 2.4. Hydroxylase and ER-Associated FA Desaturases

Approximately 14 species from 10 families have been documented to accumulate HFAs in seed oils [28,65]. In *P. lindheimeri*, PlFAH12 is one of the key enzymes responsible for HFAs synthesis (Figure 1). FAH12 shares a high degree of sequence homology and similar reaction mechanisms with ER-associated microsomal FA desaturase 2 (FAD2), which exists in all plant species [28]. FAD2 introduces a double bond at the Δ12 position of 18:1 on PC and converts it to 18:2 [37], whereas FAH12 introduces a hydroxy group at the same position of 18:1-PC, resulting in 18:1OH, which is structurally related to 18:2. In fact, lesquerella PfFAH12 is a bifunctional oleate 12-hydroxylase: desaturase [66]. We identified a full-length *PlFAH12* (c58040_g1_i2) and an almost full-length *PlFAD2* (1077 bp) (c58040_g1_i1) (Appendix A) from our transcriptomes. Gene expression analysis using quantitative PCR (qPCR) revealed that *PlFAH12* and *PlFAD2* had distinct expression profiles. *PlFAH12* had a bell-shaped pattern with a peak time at 35 DAP during seed development, and no expression was detected in the flower bud and leaf (Figure 3A). The increased expression of *PlFAH12* coincided with the increased synthesis and accumulation of HFA-containing TAG during *P. lindheimeri* seed development [15,30]. The overall expression profile of *PlFAH12* is similar to that of *PfFAH12* in lesquerella [58]. Our results are also consistent with castor *RcFAH12* expression, whose transcripts exist only in HFA-accumulating seed tissues, i.e., endosperm and embryo, and not in any other organs where no HFA is detected [49]. Therefore, organ-specific transcriptional activation of *FAH12* gene expression plays a key role in synthesizing HFA in seeds. During *P. lindheimeri* seed development, 18:2 is a major FA in young seeds (14–21 DAP), comprising 35.6–37.8% of total FA. When seeds develop to 28 DAP where HFA begins to accumulate, 18:2 drops to 22.9%, while during the remaining stages (35–56 DAP), 18:2 drops further to 8.6–5% [15]. The expression profile of *PlFAD2* showed a declining pattern during seed development (Figure 3B), coinciding with the decreased accumulation of 18:2 during *P. lindheimeri* seed development [15]. The *PlFAD2* transcript level in the leaf was similar to that of young seeds at 21 DAP, but in the flower bud it was three-fold higher (Figure 3B). To compare protein sequence similarities among *FAD2*-related genes, a full-length cDNA of *PlFAD2* was cloned by designing PCR primers based on the identified *PlFAD2* contig (Appendix A; Appendix A, method of cDNA isolation) and *PfFAD2* sequence (Genbank ID, DQ518313), since *PlFAD2* and *PfFAD2* share a high degree of sequence homology (Appendix A). The alignment of protein sequences revealed that all FAD2s and FAH12s had three identical His-boxes and six similar transmembrane domains (Figure 3C). PlFAD2 also showed seven aa residues identical to all FAD2s, but these varied among FAH12s (Figure 3C) [13,65]. The roles of these seven aa have been assessed by using site-directed mutagenesis, and the results indicate that at least four aa substitutions are required to convert an FAD2 to an FAH12, and reciprocally, six aa substitutions are required to convert an FAH12 to an FAD2 [65]. Our isolation of full-length PlFAD2 provides a new candidate for further studying the factors that determine the activity of this group enzymes.

Arabidopsis, Camelina and *Physaria* are all from Brassicaceae, whereas castor belongs to Euphorbiaceae. Similar relationships were observed from the phylogenetic analysis of protein sequences which resulted in a tree with a big clade composed of all FAD2s and FAH12s from Brassicaceae, and a separate clade containing only castor RcFAD2 and RcFAH12 (Figure 3D). PlFAD2 shared the highest aa sequences similarity with PfFAD2 at 98.5%, followed by AtFAD2 and CsFAD2s at 93.8–94.1% (Appendix A), indicating that FAD2s were highly conserved among *Physaria*, Arabidopsis and Camelina. PlFAH12 shared 90.7% aa sequence similarity with PfFAH12, followed by RcFAH12 at 62.9% (Appendix A). It was clear that all *Physaria* sequences resided in a sub-clade (grey shaded clade, Figure 3D), showing that *Physaria* FAH12s and *Physaria* FAD2s shared a common ancestor (Figure 3D). Likewise, the castor RcFAH12 also shared a common ancestor with RcFAD2 (Figure 3D). Our results of phylogenetic analysis support the hypothesis that FAH12 has evolved from the FAD2 gene through gene duplication, and such events occur independently among species during evolution [28,65].

During seed development, *P. lindheimeri* accumulates moderate levels of 18:3 in a declining pattern, showing 7–11.6% in 14–21 DAP young seeds, 7.3–2.9% in 28–42 DAP mid-mature seeds, and 1.9–2.1% in 49–56 DAP dry seeds [15]. 18:3 is synthesized through the desaturation of 18:2 by FAD3 [38]. In the *P. lindheimeri* transcriptome, we identified two contigs, c29986_g5_i1 and c37413_g1_i1, encoding PlFAD3s (Appendix A). Contig c29986_g5_i1 was a partial coding sequence and shared sequence homology with a previously identified lesquerella *PfFAD3-1* (MF611845), thus designated as *PlFAD3-1* (Appendix A). Using sequence information from *PfFAD3-1*, we cloned a full-length *PlFAD3-1* (Appendix A). Contig c37413_g1_i1 shared 93.5% nucleotide identity with MF611846, a lesquerella ortholog *PfFAD3-2* [67], therefore designated as *PlFAD3-2* (Appendix A). Using qPCR, *PlFAD3-1* and *PlFAD3-2* showed distinct expression profiles (Figure 4A,B). During seed development, *PlFAD3-1* had a bell-shaped pattern peaking at 35 DAP, while *PlFAD3-2* had a declining pattern from 21–49 DAP (Figure 4A,B). The expression level of *PlFAD3-1* was 11-fold higher in the flower bud and 1.7-fold higher in the leaf than that of 35 DAP seeds; however, *PlFAD3-2* transcripts were not detected in flower buds or in leaves (Figure 4A,B). The protein sequences of PlFAD3-1 and PlFAD3-2 shared 98.3% and 96.8% similarity with PfFAD3-1 and PfFAD3-2, respectively (Appendix A). Lesquerella PfFAD3-1 and PfFAD3-2 have been extensively characterized in the Arabidopsis FAD3-deficient mutant (*fad3-2*), which had a reduced 18:3 (from 20.0% in the wild type to 1.6% in *fad3-2*) [67]. The over-expression of PfFAD3-1 in *fad3-2* recovers 18:3 up to 29.9%; however, overexpressing PfFAD3-2 fails to recover the loss of 18:3 in the *fad3-2* mutant [67]. The PfFAD3-2 aa sequence has been examined to explain the loss of its function [67]. Interestingly, those altered aa found in PfFAD3-2 were also exhibited in PlFAD3-2 (Figure 4C). Among them, two clusters of alternations occurred in the highly conserved regions of His-Boxes (Red asterisks in Figure 4C). The first is in the second His box, where PlFAD3-2 and PfFAD3-2 altered the conserved amino acid sequences from HRTHH [68] to HKIHH and HKLHH, respectively (Figure 4C). Thr contains an uncharged polar R group, which is a critical position in the second His box for FAD3 activity. A single aa change from HRTHH to HRIHH diminishes the desaturase activity or changes the desaturase to an enzyme with a different function [69,70]. The second cluster change was in the third His-box HYCGTHVIHH in *Physaria* FAD3-2s, which was different from the known conserved region of the third His box, HDIGTHVIHH, in other FAD3s (Figure 4C) [67]. Although it is possible that these major alterations in *Physaria* FAD3-2s result in changes of enzyme activity or substrate specificity, additional in vivo and in vitro experiments are required for testing such possibilities. Our previous study also shows that PfFAD3-1 belongs to common FAD3s, whereas PfFAD3-2 has similarities with a divergent FAD3 group from the Brassicaceae family [67]. Phylogenetic analysis grouped PlFAD3-1 and PlFAD3-2 with PfFAD3-1 and PfFAD3-2, respectively (Figure 4D). The identification and isolation of PlFAD3s from this study provides new candidates for investigating the biological function and molecular evolution process among plant FAD3s.

### 2.5. Lyso-PC Acyltransferase, FA-Condensing Enzyme, and Phospholipases A

The major HFA in *P. lindheimeri* is 20:1OH, comprising 85% of seed oil. As depicted in Figure 1, the 18:1 in cytosol can be acylated to PC in the ER by the forward reaction of LPCAT, followed by hydroxylation through FAH12 to form 18:1OH-PC. The reverse reaction of LPCAT removes the 18:1OH from PC to the cytosol to be elongated to 20:1OH by KCS18. We did identify contig c56783_g1_i1 encoding a full-length *PlLPCAT* (Appendix A). The expression of *PlLPCAT* exhibited a constitutive pattern among the leaf, flower bud, and developing seeds at various stages (Figure 5A), indicating its important house-keeping function in the lipid metabolism of various organs. PlLPCAT shared 83.5% to 96.8% sequence similarities with Arabidopsis, Camelina and lesquerella, and 76.4% with RcLPCAT (Appendix A). The similarity among these LPCATs was also shown in a phylogenetic tree (Figure 5B) and protein alignment (Figure 5C). Studies of seven LPCATs from five different species, including Arabidopsis, lesquerella and castor, reveal that all seven LPCATs remove 18:1OH from PC at a three- to six-fold faster rate than acylating 18:1 to PC [24]. It is highly likely that PlLPCAT plays a critical role in removing the 18:1-OH from PC to the cytosol acyl-CoA pool.

PlKCS18 is directly responsible for elongating 18:1OH-CoA to 20:1-OH-CoA in *P. lindheimeri* (Figure 1). In our transcriptome, we identified a full-length *PlKSC18* (c61684_g3_i1) showing 82% nucleotide sequence identity with *PfKCS18* (Appendix A). Like *PlFAH12*, *PlKCS18* had a bell-shaped pattern peaked at 35 DAP during seed development, and no expression in the flower bud and leaf (Figure 6A). The nearly identical temporal and spatial expression patterns between *PlFAH12* and *PlKCS18* suggested that HFA synthesis and accumulation in *P. lindheimeri* were coordinately regulated in terms of gene expression, similarly to *PfFAH12* and *PfKCS18* in lesquerella [58]. A protein sequence comparison indicated that PlKCS18 shared 94.6% aa similarity with PfKCS18, 82.1–82.8% with AtKCS18 or CsKCS18, and 43.7% with RcKCS18 (Appendix A). The sequence similarities among KCS18 proteins were also shown in the phylogenetic tree (Figure 6B). Alignments of the aa of PlKCS18 exhibited two transmembrane domains and a region containing an active Cys site conserved among all KCS enzymes (Figure 6C) [71]. Although AtKCS18 (or AtFAE1 in some publication) exhibits activity upon elongating 18:1OH to 20:1OH [28,44,72], PfKCS18 (or LfKCS3 in some publication) is highly efficient in elongating 18:1OH in transgenic Arabidopsis [30] and Camelina [73]. The identification of PlKCS18 in this study provides another candidate for the further investigation of the mechanism of substrate recognition and specificity of the KCS family enzymes.

We depicted a PLA_2_-tpye enzyme that could release HFA at the *sn*-2 position of PC to the cytosol acyl-CoA pool (Figure 1). Based on the published PLAs known to cleave 18:1OH at the *sn*-2 of PC, including *RcsPLA_2_α* [29], *RcpPLAIIIβ* [51], *PfLCAT-PLA* or *RcLCAT-PLA* [52], we searched our transcriptomes for the orthologs in *P. lindheimeri*. We did not find *PlsPLA2α*, but we identified contigs c48215_g2_i1 and c55398_g1_i1 encoding full-length *PlpPLAIIIβ* and *PlLCAT-PLA*, respectively (Appendix A). Both transcripts were present ubiquitously in the leaf, flower bud, and developing seeds at 21, 35 and 42 DAP, indicating their house-keeping functions in *P. lindheimeri* (Appendix A. Appendix A). The mechanisms of PLA-mediated HFA cleavage and the incorporation of the HFA into seed TAG remain largely unknown. The co-expression of *RcpPLAIIIβ* and *RcFHA12* in Arabidopsis results in reductions in HFA contents in both PC and TAG, and it is suggested that the HFAs released from PC may be directed to peroxisome for degradation via β-oxidation, and are thus unavailable for TAG synthesis [51]. However, the co-expression of *PfLCAT-PLA* or *RcLCAT-PLA* with *RcFAH12* in Arabidopsis does not change the total HFA content in TAG, but instead, it increases the proportion of HFA on the *sn*-1/3 positions of TAG at the expense of HFA at the *sn*-2 of TAG [52]. The efficient utilization of HFA cleaved by PfLCAT-PLA or RcLCAT-PLA is explained by the coordinated function of a LACS which preferentially activates HFA to HFA-CoAs, and the HFA-CoAs can be acylated to *sn*-1/3 TAG by GPAT9 or DGATs through the Kennedy pathway [52]. We identified *PlLACS8* and *PlLACS9* as constitutively expressed among different organs and during seed development (Appendix A). The role of PlLACS8 and PlLACS9 in HFA-CoA formation could be investigated. It is also possible that there is an acyl-CoA-binding protein that specifically binds the HFA-CoA and transports the HFA-CoA to a correct subcellular compartment for TAG biosynthesis [52]. In *P. lindheimeri* seed cytosol, PlKCS18 is expected to efficiently elongate 18:1OH-CoA to 20:1OH-CoA; therefore, PlKCS18 should also be a key factor in this PLA-mediated generation of 20:1OH-CoA. The *PlpPLAIIIβ* and *PlLCAT-PLA* identified in this study provide candidates for the future investigation of the role of PLA-type enzymes in HFA-TAG synthesis in *P. lindheimeri*.

### 2.6. Kennedy Pathway for TAG Synthesis in the ER

The Kennedy pathway for TAG synthesis utilizes three acyl-CoA-dependent acyltransferases, GPAT, LPAT and DGAT, that sequentially acylate the *sn*-1- and *sn*-2- and then the *sn*-3-position of G3P with acyl-CoA (Figure 1, blue arrows). Since the synthesis of membrane glycerolipids also begins with the sequential acylation of the *sn*-1- and *sn*-2-positions of G3P, GPAT and LPAT are common to the synthesis of TAG and membrane glycerolipids. Arabidopsis GPAT9 (At5g60620) has been demonstrated to participate in seed oil and membrane lipids biosynthesis [74,75]. Using At5g60620 as a query, we identified a contig, c58129_g2_i1, encoding a full-length *PlGPAT9* (Appendix A). Gene expression profiling indicated that *PlGPAT9* was expressed in various organs, showing a bell-shaped temporal pattern peaking at 35 DAP (Figure 7A). The *PlGPAT9* transcript level in the leaf was similar to that of the 35 DAP seeds, and the level in the flower bud was two-fold higher than that in the leaf (Figure 7A). Phylogenetic analysis showed a close relationship between PlGPAT9 and PfGPAT9 (Figure 7B). The PlGPAT9 protein sequence shared 85.4% aa similarity with PfGPAT9, which has 1–51 aa deletion at the N-terminal (Figure 7C, Appendix A). When the first 51 aa were removed from PlGPAT9, the remaining aa similarity between PlGAPT9 and PfGPAT9 reached 98.8% (Appendix A). Compared with GPAT9s from Arabidopsis, Camelina, and castor, PlDGAT9 showed similarities from 88.1 to 95.8% (Appendix A). High protein sequence similarity among GPAT9 proteins indicates that GPAT9s have evolved conservatively. Most GPAT members have a broad acyl-CoA substrate specificity [25,76], but limited studies have been conducted to understand the role of GPAT9 in HFA-TAG biosynthesis. Over-expressing castor *RcGPAT9* alone does not change the HFAs content in transgenic Arabidopsis CL37 [48,50], which accumulates a background of 17% HFAs in seed TAG due to the expression of *RcFAH12* [28]. This is in contrast with over-expressing single castor gene, *RcLPAT2* [48,49], *RcDGAT2* [46], and *RcPDAT1A* or *RcPDAT1-2* [35,36], where increased HFAs production has been observed in CL37. However, when *RcGPAT9* is co-expressed with *RcLPAT2* and *RcDGAT2* in CL37, seed TAG contains more HFAs, and besides this, most HFAs are acylated in the *sn*-1/3 position compared with the line only co-expressing of *RcLPAT2* and *RcDGAT2* [50]. In addition, co-expressing *RcGPAT9* with *RcLPAT2* and *RcPDAT1A* increases not only total HFAs but also substantial amounts of tri-HFA-TAG (TAG species with all three *sn* positions occupied with HFAs) in CL37 seeds, whereas in the line co-expressing only *RcLPAT2* and *RcPDAT1A*, there is no tri-HFA-TAG detected [50]. These results suggest that RcGPAT9 plays an important role in acylating HFAs at the *sn*-1 position of G3P, resulting in *sn*-1-HFA-LPA, which facilitates the subsequent incorporation of *sn*-2 and *sn*-3 HFA into seed TAG by *sn*-2- and *sn*-3-specific acyltransferases. It would be interesting to investigate if PlGPAT9 plays similar role to RcGPAT9 in HFA-TAG synthesis. Sequence alignment showed that in the second transmembrane domain, one aa at the 105th position of PlDGAT9 was Thr (Figure 7C). The corresponding aa in lesquerella and castor GPAT9s was also Thr, but for Arabidopsis and Camelina it was Ala (Figure 7C). The further examination of 39 plant GPAT9s from 30 species and 11 families revealed that the Thr^105^ was predominantly present in GPAT9s from Fabaceae, Euphorbiaceae, Ranunculaceae, Phrymaceae, Malvaceae, Solanaceae and Rutaceae, while the Ala was predominantly present in GPAT9s from Brassicaceae and Poaceae (Appendix A). It was intriguing that among 11 Brassicaceae GPAT9s, only *Physaria* GPAT9s had the Thr corresponding to the 105th position in PlDGAT9 (Appendix A). Thr and Ala are neutral aa, but Thr is polar, whereas Ala is nonpolar. Whether the difference in hydrophobicity between Thr and Ala affects GPAT9 substrate selection remains to be investigated.

LPAT acylates the *sn*-2 position on LPA with an acyl-CoA to produce the PA precursor of membrane lipids and TAG. In most seed TAG synthesis pathways, this reaction is typically catalyzed by an LPAT2 that has high substrate specificity for C_18_ unsaturated FAs (i.e., 18:1, 18:2, and 18:3) [77]. Using Arabidopsis *LPAT2* (At3g57650) as query, we identified *PlLPAT2* (c53827_g2_i7, Appendix A). *PlLPAT2* expressed constitutively during seed development, and the level was 1.2-fold higher in the leaf and 5-fold higher in the flower bud than that of 21 DAP seeds (Figure 8A). Our spatial and temporal expression profiles of *PlLPAT2* indicate that *PlLPAT2* was likely involved in membrane lipids and storage TAG biosynthesis. The results of protein sequence analysis indicate that PlLPAT2 and PfLPAT2 share 95.2% aa similarity (Appendix A). Phylogenetic analysis grouped PlLPAT2 with PfPLPAT2 (Figure 8B). Despite sequence similarity, PlLPAT2 may differ from PfLPAT2 in acyl-CoA substrate specificity. Lesquerella TAGs contain ~60% 20:1OH and almost all of it is esterified to the *sn*-1 and *sn*-3 positions, while the *sn*-2 positions are exclusively occupied with C_18_ unsaturated FAs [54]. The *sn*-2 FA profile of lesquerella TAG implies that PfLPAT2 may act like a typical plant LPAT2 with a high substrate specificity to C_18_ unsaturated FAs [77]. In *P. lindheimeri*, seed TAGs contain 85% 20:1OH, with most of them having 20:1OH at all three *sn*-positions, similar to castor oil, where over 70% are tri-18:1OH-TAGs [53]. PlLPAT2 probably plays a key role in acylating 20:1-OH-LPA during seed development in *P. lindheimeri*. The alignment of protein sequences revealed that PlLPAT2 shared two aa, Asn^230^ (neutral, polar) and Leu^255^ (neutral, non-polar) with RcLPAT2 (indicated by red asterisks and boxes in Figure 8C), whereas the corresponding aa among PfLPAT2, AtLPAT2 and CsLPAT2 were Thr and Ser (both neutral, polar) (Figure 8C). To further examine the differences, 32 available LPAT2s from plants were aligned (Appendix A). Although the Leu^255^ was present in 15 sequences representing 11 families, the combination of Asn^230^ and Leu^255^ was conserved in *P. lindheimeri* and the species from Euphorbiaceae, including castor, Tung tree (*Vernicia fordii*) and Jatropha (*Jatropha curcas*) (Appendix A). Like *P. lindheimeri*, castor and Tung tree seed oils contain over 80% unusual FAs, ricinoleic acid and eleostearic acid (conjugated FA, 18:3Δ9cis,11cis,13trans, or ESA), respectively [78]. The ESA in Tung oil is synthesized by an FADX, a conjugase X also derived from FAD2 [79]. Although the major FAs in Jatropha oil are 18:1 and 18:2, there are 1–2% of ESAs in seed TAG [80], suggesting that Jatropha seeds have a weak conjugase activity. The possibility exists that Jatropha acyltransferases, including JcLPAT2, acylate ESA into TAG. More experiments are required to investigate the substrate specificity of PlLPAT2, RcLPAT2, VfLPAT2 and JcLPAT2, and to examine the role of Asn^230^ and Leu^255^.

Most plant species contain five Class-A LPATs, LPAT1 to LPAT5 [77,81,82]. LPAT1 is the only member localized in plastids showing high substrate specificity for palmitic acid (16:0)-CoA and 18:0-CoA [81,83]. The rest of the LPATs are ER-localized [82]. In some seed oils accumulating unusual FAs, there are additional Class-B LPATs. Class-B LPATs are often expressed in seeds and are associated with the acylation of unusual acyl-CoAs for TAGs, such as erucic acid (22:1) in Limnanthes [84], lauric acid (12:0) in Cocos [85], and caprylic (8:0) and capric acid (10:0) in Cuphea [86]. *LPAT-B* genes have been used to improve the unusual FA content in other oilseeds by genetic engineering [86,87,88,89]. Seven castor LPATs (RcLPATs) have been identified [49,90], and all of them encode functional enzymes [49]. In vitro enzyme assays have indicated that RcLPAT2 has a strong preference for 18:1OH-CoA when *sn*-1-18:1OH-LPA was used as an acyl acceptor; however, when *sn*-1-18:1-LPA was used as an acyl acceptor, RcLPAT2 selected 18:1-CoA [90]. The phenomenon of RcLPAT2 having different specificities toward acyl-donor and acyl-acceptor is supported by in vivo analysis of TAG species in transgenic lesquerella lines expressing *RcLPAT2* [55]. Interestingly, *RcLPAT-B* is expressed in diverse organs [49,90], and exhibits broad substrate specificity, with saturated CoAs (12:0–16:0) being the preferred substrates [90]. When all seven *RcLPATs* are expressed in an HFA-producing Arabidopsis CL37, *RcLPAT2, RcLPAT3A*, and *RcLPATB* increase the HFAs content in Arabidopsis [49]. Increasing evidence indicates that unusual FAs are acylated to TAG by variant LPATs [48,86,91,92]. In our transcriptomes, we identified *PlLPAT1* to *PlLPAT5* (Appendix A). *PlLPAT2* transcript levels were much higher than those of *PlLPAT3*, *PlLPAT4* and *PlLPAT5* in the developing seed, leaf and flower bud (Appendix A). We did not find orthologs of *RcLPAT-*B and *RcLPAT3A* in *P. linheimeri* transcriptomes. *PlLPAT2* likely is the isoform for acylating 20:1OH at the *sn*-2 position of 20:1OH-LPA. Further studies, such as in vitro enzyme assays and the in vivo transgenic expression of *PlLPAT2* in lesquerella or other oilseeds, would determine its role in HFA-TAG synthesis.

DGAT catalyzes the final step of TAG biosynthesis in the Kennedy pathway (Figure 1). There are three sequence-unrelated classes of DGATs reported in plants: membrane-bound DGAT1 and DGAT2, and cytosolic DGAT3 [93]. Using Arabidopsis *DGAT1* (At2g19450), *DGAT2* (At3g51520), and *DGAT3* (At1g48300) as queries, we identified contigs, c56288_g1_i1, c47089_g1_i3, and c56599_g1_i1, encoding PlDGAT1, PlDGAT2, and PlDGAT3, respectively (Appendix A). All three *PlDGATs* were ubiquity expressed in the developing seed, flower buds and leaf (Figure 9A,C,E), indicating that PlDGATs acted at least in part in the general TAG metabolism in *P. lindheimeri*. Phylogenetic analyses indicated that *P. lindheimeri* DGATs were closely related to their lesquerella orthologs (Figure 9B,D,F) (Figure 10A–C), showing aa similarity at 84.3% with PfDGAT1-1, 84.9% with PfDGAT2, and 89.4% with PfDGAT3, respectively (Appendix A). Among these three DGAT families, the role of DGAT3 in seed TAG synthesis has not yet been established [94], but it has been shown to be involved in recycling FAs in general lipid metabolism [94,95,96]. DGAT1 and DGAT2 have been intensively studied. Large amounts of evidence indicate that DGAT1 is a major determinant in seed oil accumulation in most species studied [93,97,98,99]. Recent studies using enzyme assays suggest that different DGAT families have different specificities toward acyl-donor and acyl-acceptor. Camelina CsDGAT1 excludes, whereas CsDGAT2 selects, acyl-acceptors containing only polyunsaturated FAs [100]. In *Brassica napus*, DGAT1 variants prefer acyl-donors with saturated (e.g., 16:0-CoA) and monounsaturated (e.g., 18:1-CoA) acyl-donors, whereas DGAT2 variants strongly select 18:3-CoA [101]. In plant seeds accumulating unusual FAs, DGAT2s are essential enzymes in acylating unusual *FAs* to the *sn*-3 position of DAG. For example, castor RcDGAT2 prefers 18:1OH to common FAs [46,102]. The *RcDGAT2* transcript is expressed 18-fold higher than that of *RcDGAT1* in developing seeds, supporting the role of RcDGAT2 in acylating 18:1OH-CoA to DAG [102]. Tung tree VfDGAT2 selects ESA [48,103]. It seems that tung tree DGAT1 and DGAT2 are localized in different regions of the ER, and consequently they may select different substrates [103]. *Brassica napus* DAGT2 variants, BnaA.DGAT2.d and BnaA.DGAT2.e, select not only 18:3 but also unusual erucic acid (22:1) [101]. In some species, isoforms of DGAT1 and DGAT2 are both capable of acylating unusual FA, such as vernolic acid (cis-12,13-epoxy–cis-9-octadecenoic, 18:1>O) by VgDGAT1 and VgDGAT2 from ironweed (*Vernonia galamensis*) [104], and 22:1 by ChDGAT1 and ChDGAT2 from Crambe (*Crambe hispanica*) [105]. It is worth noting that *Euonymus alatus* diacylglycerol acetyltransferase (EaDAcT) acylates acetyl-CoA to the *sn*-3 position of DAG, and it belongs to the membrane-bound O-acyltransferase (MBOAT) super-family, which also includes DGAT1 [106,107]. Nevertheless, it is clear that castor RcDGAT2, but not RcDGAT1, acylates 18:1OH both in vitro [102] and in vivo [46].

Considering the similarities between 18:1OH and 20:1OH, it is likely that PlDGAT2 plays a key role in assembling seed TAG in *P. lindheimeri*. No detailed structural analysis of DGAT2 has been reported, although various motifs have been proposed to serve as important binding or active sites [93,98,99]. Protein sequence alignment of DGAT2s from non-HFA-producing plants, Arabidopsis and Camelina, as well as HFA-producing plants, castor, lesquerella and *P. lindheimeri* (Figure 10B), showed the following features: (1) castor RcDGAT2 had an extra 25 aa at the N-terminal of the protein, and nine Asn residues were located from the 6th to the 18th positions; (2) *P. lindheimeri* also had 25 extra aa (position 41th to 75th), but these were located in the second transmembrane domain; (3) three aa, Glu (polar, negative), His (polar, positive) and Lys (polar, positive), at the 3rd, 115th, and 212th positions of PlDGAT2, were conserved among HFA-producing plants, while the corresponding positions are Gly (nonpolar, neutral), Tyr (polar, neutral), and Gln (polar, neutral), respectively, in Arabidopsis and Camelina. Various conserved motifs have been proposed to serve as important binding or active sites for DGAT2s [93,98,99]. In vitro analysis reveals that the N-terminal region (the first 30–50 aa) of mouse DGAT2 or yeast DGAT2 is not essential for DGAT activity in vitro [108,109]. The alignment of 32 DGAT2s from different plant species showed that the N-terminals of DGAT2s were highly variable in length (Appendix A), thus it was unlikely that the extra 25 aa at the N-terminal of RcDGAT2 was critical for HFA-TAG assembly. The 3rd aa also varied among different DGAT2s capable of acylating unusual FAs (Appendix A). Although the Glu^3^ is found in ironweed VgDGAT2 and Crambe ChDGAT2s, which acylate unusual epoxy vernolic acid FA (18:1 > O) [104] and 22:1 [105], respectively, different aa Met^3^ (non-polar, neutral) and Lys^3^ are located in tung tree VfDGAT2 and rapeseed BnaA.DGAT2.d and BnaA.DGAT2.e (Appendix A), which also select unusual FAs. VfDGAT2 has been demonstrated to acylates ESA [48,103], whereas both BnaA.DGAT2.d and BnaA.DGAT2.e select 18:3 and 22:1 in high-erucic acid MAPLUS cultivar [101]. Thus the 3^rd^ aa of DGAT2s is not critical for DGAT2s in selecting unusual FAs. Among the 38 plant DGAT2s, PlDGAT2 was the only one having the additional 25 aa in the second transmembrane domain (Appendix A). It is doubtful that these extra aa are essential for PlDGAT2 activity, but further studies are required to determine the relationship between the structure and function of PlDGAT2. Two motifs, YEP and PH-block (indicated by blue boxes in Figure 10B), are found to be essential for DGAT2 function in mouse and yeast [108,109]. The His^115^ (red asterisk in Figure 10B) located between the YEP and PH-block is conserved among 23 DGAT2s from a broad range of families, including Poaceae, Euphorbiaceae, Celastraceae, Brassicaceae, and Fabaceae (Appendix A), thus it is unlikely that the His^115^ is a specific aa for DGAT2s in selecting unusual FAs. The Lys^212^ is located inside a conserved GGE-block (indicated by a blue box in Figure 10B) whose function has not yet been identified [98]. It is intriguing that 11 out of 35 of the DGAT2s have corresponding Lys (Appendix A), and eight of them are from species accumulating various unusual FAs, such as 18:1OH in castor, 20:1OH in lesquerella and *P. lindheimeri*, ESA in tung, and 22:1 in crambe. The remaining three have substrate preferences for polyunsaturated FA, including rapeseed BnaA.DGAT2.2b and BnaA.DGAT2.2c for selecting 18:3 [101], and Jatropha JcDGAT2 for selecting 18:2 [110]. Future experiments are required to determine the importance of the Lys^212^ in PlDGAT2 and other DGAT2s in conducting enzyme activity.

### 2.7. Acyl-CoA-Independent and PC-Mediated Pathways for TAG Synthesis

Unlike DGAT, which utilizes the acyl-CoA as acyl donor, PDAT transfers the acyl moiety at the *sn*-2 position of PC to the *sn*-3 position of DAG, yielding TAG [34,111] (Figure 1). Using Arabidopsis *AtPDAT1* (AT5G13640.1) and *AtPDAT2* (AT3G44830.1) as queries, we identified c61686_g2_i4 and c62019_g1_i1 encoding *PlPDAT1* and *PlPDAT2*, respectively (Appendix A). Gene expression analysis showed that the *PlPDAT1* transcript level was high in the 35 DAP seed and flower bud, moderate in 21, 28 and 42 DAP seeds and leaf, but not detectable in 49 DAP desiccating seed (Figure 11A). *PlPDAT2* expressed ubiquitously during seed development as well as in the flower bud and leaf (Figure 12A). Almost all plant species contain at least PDAT1, thus PDAT1s have been characterized more extensively than PDAT2s. In Arabidopsis, AtPDAT1 plays an essential role in TAG biosynthesis in developing seeds and pollen [112,113,114] for maintaining membrane lipid homeostasis through directing the FA toward β-oxidation [115], and in stress responses [116]. Similar to AtPDAT1, Camelina CsPDAT1 also participates in stress responses [117]. The role of CsPDAT1 in controlling seed oil content remains controversial. A mutation in *CsPDAT1* reduces oil content [118], and either up-regulating or down-regulating *CsPDAT1* does not change seed oil content [118,119]. Studies indicate that CsPDAT1 has a substrate preference for polyunsaturated FAs, 18:2 and 18:3 [100,117,118,119]. Similar results are also reported for *AtPDAT1* [111], sunflower *PDAT1* [120] and flax PDATs (*LuPDAT1* and *LuPDAT5*) [121], with the exception of sesame (*Sesamum indicum*), where the over-expression of *SiPDAT1* in yeast H1246 results in increased 18:1 and 18:2 levels, but not 18:3 [122]. The role of *PlPDAT1* and *PlPDAT2* remains to be determined. Based on the spatial and temporal expression patterns of *PlPDAT1* (Figure 11A) and *PlPDAT2* (Figure 12A), *PlPDAT1* might play an important role in TAG synthesis, as it was expressed at higher levels in developing seeds than *PlPDAT2*. The ubiquitous expression patterns of *PlPDAT2* suggest its house-keeping function for maintaining normal plant growth and development. The results of protein sequence comparison showed that PlPDAT1 shared higher aa similarity with AtPDAT1 (92.1%) and CsPDAT1s (91.5–92.1%) than with RcPDAT1 (74.7%) (Figure 11B,C) (Appendix A). PlPDAT2 displayed 86.2–86.2% aa similarities with AtPDAT2 and CsPDAT2, and 60.5% with RcPDAT2 (Figure 12B,C) (Appendix A). We excluded PfPDATs from protein sequence analysis, because PfPDATs were not full-length. Castor has two PDAT1s, RcPDAT1-1 (or RcPDAT1B) and RcPDAT1-2 (or RcPDAT1A) [35,36]. RcPDAT1-2 (RcPDAT1A), not RcPDAT1-1 (RcPDAT1B), is demonstrated to be the isoform having substrate preference for 18:1OH-PC, and it participates in TAG synthesis [35,36]. Castor *RcPDAT2*, Arabidopsis *AtPDAT2*, and flax *PDAT2s* (*LuPDAT3* and *LuPDAT6*) do not show an apparent function in TAG biosynthesis [35,36,111,121]. Considering the similarly high HFA contents in castor and *P. lindheimeri*, PlPDAT1 may play a similar role to RcLPAT1-2 (RcLPAT1A) in trans-acylating 18:1OH from *sn*-2 PC to *sn*-3 DAG for TAG formation. More detailed examinations of PDAT1 sequences revealed that PlPDAT1 had the typical conserved features of lecithin:cholesterol acyltransferase LCAT enzymes [123,124,125], including the ‘lid’ region between two cysteines with a tryptophan in the middle, a salt bridge, and a catalytic triad with a set of three coordinated amino acids S-D-H (or Ser-Asp-His) in the active site of acyltransferase [126] (Figure 11C). The Thr and Asp corresponding to the aa at the 390th and 415th positions of PlPDAT1 were conserved between PlPDAT1 and RcPDAT1-2, but varied among AtPDAT1, CsPDAT1s, and RcPDAT1-1 (Figure 11C). The further alignment of 32 plant PDAT1s showed that Thr^390^ was present in 10 PDATs from four different families—Brassicaceae, Malvaceae, Ranunculaceae, and Euphorbiaceae (Appendix A). Asp^415^ was conserved among PlPDAT1, RcPDAT1-2, and two *Citrus* PDAT1s (Appendix A). It seems unlikely that either Thu^390^ or Asp^415^ alone would be indispensable to determine the selectivity of HFAs by PlPDAT1 or RcPDAT1-2. However, Thu^390^ and Asp^415^ were close to each other (25 aa apart), located in the catalytic trial domain, and the combination of Thu^390^ and Asp^415^ was found only in PlPDAT1 and RcPDAT1-2, so it is possible that the combination of Thu^390^ and Asp^415^ in PlPDAT1 and RcPDAT1-2 could play a role in catalytic site geometry for determining the enzyme specificity of PlPDAT1 and RcPDAT1-2 in trans-acylating HFA-PC. Future experiments are required to verify this possibility. The isolation of *PlPDAT1* will be of value in studying the factors that determine the HFA-preferring substrate of plant PDATs.

PDCT catalyzes the interconversion between DAG and PC through the transfer of the phosphocholine group from PC to DAG. In *P. lindheimeri*, acyl editing and PC-DAG interconversion through PlLPCAT and PlPDCT, respectively, may co-contribute to the formation of TAGs with enriched 20:1OH (Figure 1). In *P. lindheimeri* transcriptomes, we identified one contig, c58276_g1_i1, showing 85% nucleotide identity with Arabidopsis *PDCT*, and thus designated it as *PlPDCT* (Appendix A). Gene expression analysis indicated that *PlPDCT* was expressed ubiquitously in developing seeds at various stages, in flower bud and in leaf tissues (Figure 13A). The PlPDCT protein sequence shared 73.1–86.4% with AtPDCT and CsPDCTs, and 61.8% with RcPDCT (Appendix A, Figure 13B). AtPDCT [41] and CsPDCT [100] are not able to use HFAs substrates, whereas in species where the seed oils contain high levels of unusual FAs, such as 18:1OH in castor and cyclopropane FAs (CPAs) in lychee (*Litchi chinensis*), their PDCTs have been shown to facilitate the efficient accumulation of HFAs in Arabidopsis [41] and CPAs in Camelina [127]. The alignment of the PlPDCT with AtPDCT, CsPDCT, and RcPDCT revealed a central core of high aa similarity (Figure 13C), showing a catalytic triad with a set of three coordinated amino acids, H-H-D (or His-His-Asp), in the conserved C2 and C3 domains of the lipid phosphatase/phosphotransferase family [128]. In addition, PlPDCT had Leu, Lys, and Leu at the 40th, 71th and 295th positions that were identical to the aa aligned with RcPDCT (Figure 13C), and the corresponding aa of AtPDCT and CsPDCTs were Arg, Thr, and Val, respectively (Figure 13C). The PfPDCT sequence was partial, so it was not included in the comparison. To further examine if the Leu^40^, Lys^71^, and Leu^295^ were unique to PlPDCT and RcPDCT, 28 plant PDCTs were aligned (Appendix A). The Leu^40^ and Leu^295^ were located in the N- and C-terminal regions that were highly variable in length and aa identity (Appendix A), thus the Leu^40^ and Leu^295^ were considered as conserved aa for neither PlPDCT nor for RcPDCT. The Lys^71^ was present in 9 out of 28 PDATs representing six different families (Appendix A). Therefore, it was also unlikely that the Lys^71^ played an essential role for RcPDCT or for PlPDCT in HFA-PC-mediated TAG synthesis. Using a transgenic approach [41] and flux labeling analysis [129] should help to determine the role of PlPDCT in channeling HFAs into seed TAG. In addition to PDCT, plant phospholipase C (PLC) hydrolyzes membrane phospholipids, including PC, yielding DAG and phosphorylated head-groups. HFAs esterified at the *sn*-2 position of PCs are then converted to *sn*-2-HFA-DAGs to be utilized by PDAT and DGAT (Figure 1). A phospholipase C-like protein (*RcPLCL1*) from castor increases the amount of HFAs when co-expressed with *RcFAH12* in transgenic Camelina seed [47]. In our transcriptomes, we did not find an ortholog sequence encoding a putative *PlPLCL1*. Besides, two *Arabidopsis phospholipase D_ζ_* genes (*AtPLD_ζ1_* and *AtPLD_ζ2_*) have been recently isolated and found to increase seed TAG content in Camelina through PC to DAG conversion [130] (Figure 1). We found two contigs, c53747_g1_i1 and c55834_g1_i1, representing *PlPLD_ζ1_* and *PlPLD_ζ2_*, respectively (Appendix A). Both *PlPLD_ζ1_* and *PlPLD_ζ2_*, had constitutive spatial and temporal expression profiles, indicating their house-keeping function in *P. lindheimeri* (Appendix A). The finding of *PlPLD_ζ1_* and *PlPLD_ζ2_*, provides targets for the future investigation of their roles in HFA accumulation in seed TAGs.

## 3. Materials and Methods

### 3.1. Plant Materials and RNA Extraction

The *P. lindheimeri* seeds (accession number PI 643174) were obtained from United States Department of Agriculture, National Plant Germplasm System (http://www.ars-grin.gov/npgs/). Ten plants were grown in a greenhouse under a condition as described [131]. Developing seeds from 21, 35 and 42 days after pollination (DAP) were generated using the method as described [131]. Unopened mature flower buds, fully expended young leaves, and developing seeds were pooled from ten plants for each sample and frozen immediately in liquid nitrogen after harvest and stored at −80 °C. Total RNA was prepared using RNeasy Plant Mini Kit (Qiagen, Valencia, CA, USA).

### 3.2. cDNA Library Construction

To generate an mRNA-focused sequencing library, the mRNA from a total RNA sample was converted into a cDNA library using the reagents and instruction was provided by the Illumina^®^ TruSeq^™^ RNA Sample Preparation Kit (Illumina, San Diego, CA, USA). The first step in the workflow involves purifying the poly-A-containing mRNA molecules using poly-T oligo-attached magnetic beads. Following the purification, the mRNA is fragmented into small pieces using divalent cations under elevated temperatures. The cleaved RNA fragments are copied into first strand cDNA using reverse transcriptase and random primers. This is followed by second strand cDNA synthesis using DNA Polymerase I and RNase H. These cDNA fragments then go through an end repair process, the addition of a single ‘A’ base, and then ligation of the adapters. The products are then purified and enriched with PCR to create the final cDNA library.

### 3.3. Sequencing and Sequence Quality Check

Illumina utilizes a unique “bridged” amplification reaction that occurs on the surface of the flow cell. A flow cell containing millions of unique clusters is loaded into the HiSeq 2000 (Illumina) for automated cycles of extension and imaging. Solexa’s Sequencing-by-Synthesis (Illumina) utilizes four proprietary nucleotides possessing reversible fluorophore and termination properties. Each sequencing cycle occurs in the presence of all four nucleotides leading to higher accuracy than methods where only one nucleotide is present in the reaction mix at a time. This cycle is repeated, one base at a time, generating a series of images each representing a single base extension at a specific cluster.

FastQC (http://www.bioinformatics.babraham.ac.uk/projects/fastqc) was applied to check the quality of raw sequence data obtained from high-throughput sequencing pipelines. Trimmomatic (0.32) was used to remove adapters which are undesirable for downstream application [132].

The sequence raw data from this study have been submitted to the NCBI Sequence Read Archive (SRA) (http://www.ncbi.nlm.nih.gov/), under the BioProject: PRJNA672105.

### 3.4. Transcriptome Analysis

RNA-seq data were subjected to de novo transcriptome assembly using Trinity (r20140717) [133], which also incorporates RSEM (RNA-Seq by Expectation-Maximization) algorithm (1.2.15) for abundance estimation. RSEM is particularly useful for the quantification of de novo transcriptome assemblies when a reference genome is not available, such as the case of *P. lindhemeri* in this study. The gene expression levels (transcripts or contigs levels) were calculated as fragments per kilobase of transcript per million mapped reads (FPKM) [61]. Contigs with an FPKM value of zero in at least one sample of all five samples were excluded to eliminate transcripts (contigs) with extremely low coverage and to minimize bias of statistical analysis. Logarithm (based 2) (fold-change) was calculated for filtered contigs to define significantly differentially expressed genes (DEGs). If the FPKM value was between 0 and 1, it was converted to a negative value during log transformation. Therefore, log2-based transformation was performed on Raw Signal (FPKM) +1 value. Furthermore, quantile normalization [134] was performed to reduce the systematic/technique bias that may affect data presentation. After data scaling with logarithm and quantile normalization, a heatmap plot was generated using the “pheatmap” package in the R program [135].

### 3.5. Protein Sequence Analysis

NCBI ORF-finder (http://www.ncbi.nlm.nih.gov/orffinder/) was applied to find the open reading frame (ORF) of a nucleotide sequence from transcriptome analysis, and the resulting ORF or deduced amino acid sequence was confirmed by NCBI BLAST program (http://www.ncbi.nlm.nih.gov/BLAST/) for its protein function. Putative transmembrane domains (TM) were predicted by the TMHMM web tool (http://www.cbs.dtu.dk/services/TMHMM/). Protein sequences were aligned using the ClustalW method of MEGA7: Molecular Evolutionary Genetics Analysis version 7.0 for bigger datasets [136]. Aligned sequences were converted in to fasta format for generating a phylogenetic tree. The maximum likelihood statistical method was used and replicated 100 times with bootstrapping. The numbers shown next to the branches indicate repeated times of replicated trees clustered together during the bootstrap test.

### 3.6. Quantitative PCR

Total RNA was reverse transcribed using the QuantiTect Reverse Transcription Kit (QIAGEN, Valencia, CA, USA) according to the manufacturer’s directions. The resulting cDNA samples were used in PCR reactions. Standard PCR amplification reactions were carried out as described [58]. PCR product specificity was confirmed by melting-curve analysis and by electrophoresis on 4% agarose gel to ensure that PCR reactions were free of primer dimers and non-specific amplicons. Information of primer pairs and their PCR efficiencies is listed in Appendix A. The method of Pfaffl [137] was applied to calculate comparative expression levels between samples. The *P. lindheimeri* 18S gene was used as internal reference to normalize the relative amounts of RNAs for all samples.

## 4. Conclusions

To develop alternative oilseed crops for safe sources of HFAs production, we have built *P. lindheimeri* transcriptomes from developing seeds, leaf and flower bud samples that provide a valuable reference for discovering genes involved in the synthesis of TAGs enriched with HFAs. Clean reads were assembled into 129,633 contigs which represented 85,948 unique genes. GO analysis indicated that 60% of contigs matched genes with known functions. The comparison of DEGs revealed that the gene expression profiles in developing seeds were distinctly different from those of the leaf and flower bud. Using well-defined Arabidopsis lipid genes as queries, we identified 40 contigs, with 37 of them showing 78–100% high nucleotide sequence identity with Arabidopsis genes, which permits the translating of gene function from Arabidopsis to *P. lindheimeri*. Based on current knowledge, we proposed a pathways and characterized 16 essential *P. lindheimeri* lipid genes involved in 20:1OH-TAG biosynthesis in the seed. Manual annotation revealed that 14 out of these 16 essential genes encode full-length proteins, suggesting that the clean reads had good coverage of the assembled contigs. Detailed spatial and temporal expression profiling of these transcripts revealed various patterns. Two genes, *PlFAH12* and *PlKCS18,* directly responsible for 20:1OH synthesis were expressed only in developing seeds, and showed high peaks at 35 DAP, coinciding with the maximum 20:1OH accumulation rate during seed development. As such, the transcriptional up-regulation of *PlFAH12* and *PlKCS18* should play a key role in synthesizing 20:1OH. To understand how *P. lindheimeri* evolves to channel 20:1OH into seed TAG at a high level, the *P lindheimeri* protein sequences of enzymes in the Kennedy pathway (PlGPAT9, PlLAPT2, and PlDGAT2) and PC-mediated pathway (PlPDAT1 and PlPDCT) were compared with 28–39 correspondent orthologs from a published database. We identified the single or multiple aa conserved in each enzyme among species accumulating HFA-TAG. Our transcriptome data itself are insufficient to deduce the regulatory mechanism of HFA synthesis and accumulation in seeds. Additional studies such as genome-wide proteotypes analyses [138] and genome-wide metabolic flux analyses [139] are also essential to enhance our understanding of HFA metabolism. Nevertheless, the information obtained from this study provides a valuable resource not only for studying the mechanisms of how HFAs are accumulated, but also for the biotechnological production of HFAs, including health beneficial SHFAs in existing oilseed crops.

## Figures and Tables

**Figure 1 ijms-22-00514-f001:**
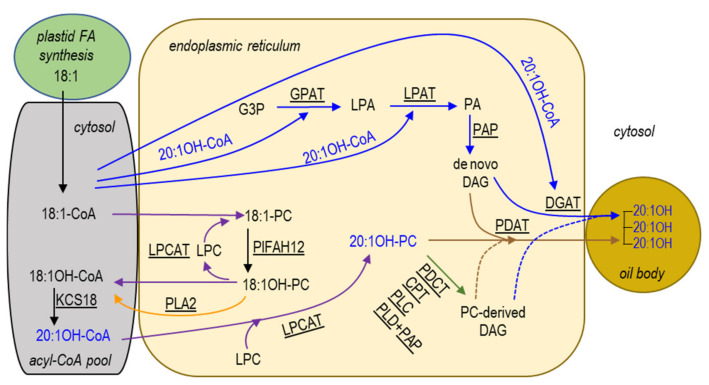
Proposed pathways for triacylglycerol (TAG) biosynthesis in *P. lindheimeri* seed. Blue arrows indicate reactions involved in Kennedy pathway. Purple arrows indicate reactions involved in acyl editing by LPCAT. Brown arrows indicate PDAT-mediated pathways. Green arrows indicate reactions involved in PC-derived DAG synthesis. Dotted lines indicate PC-derived DAG utilized by DGAT and PDAT. Enzymes catalyzing these reactions are underlined. FA numerical symbols: 18:1, oleic acid; 18:1OH, ricinoleic acid; 20:1OH, lesquerolic acid. Abbreviations: CoA, co-enzyme A; PC, phosphatidylcholine; LPCAT, lysophosphatidylcholine acyltransferase; LPC, lysophosphatidylcholine; PLA2, phospholipase A; PlFAH12, *P. lindheimeri* Δ12 oleic acid hydroxylase; KCS18 (or KCS3), 3-ketoacyl-CoA synthase 18; G3P, glycerol-3-phosphate; LPA, lysophosphatidic acid; PA, phosphatidic acid; DAG, diacylglycerol; GPAT, glycerol 3-phosphate acyltransferase; LPAT, lysophosphatidic acid acyltransferase; PAP, phosphatidic acid phosphatase; DGAT, diacylglycerol acyltransferase; CPT, CDP-choline:DAG cholinephosphotransferase; PDAT, phospholipid:DAG acyltransferase; PDCT, PC:DAG cholinephosphotransferase; CPT, CDP-choline:DAG cholinephosphotransferase; PLC, phospholipase C; PLD, phospholipase D; TAG, triacylglycerol.

**Figure 2 ijms-22-00514-f002:**
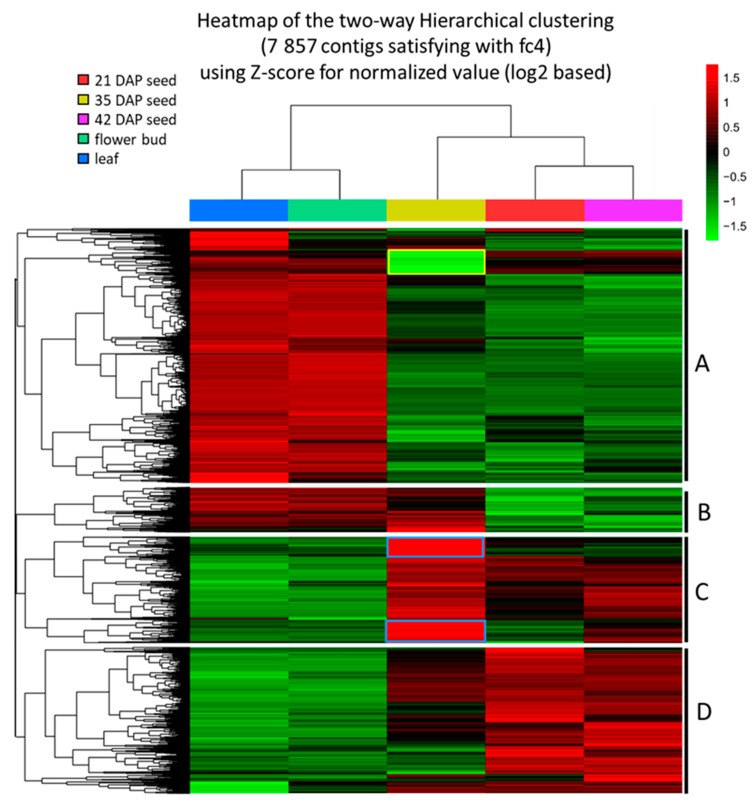
Heat map for hierarchical clustering. For a significant total of 7857 contigs, samples and contigs with similar expression levels were grouped by hierarchical clustering analysis (Euclidean method, complete linkage) using expression level (normalized value) for each sample and contigs. In the 35 DAP sample, yellow or blue boxes indicate that most of the contigs were highly down-regulated (green color) or up-regulated (red color). Groups A, B, C and D were visually divided. fc4, fold change ≥4.

**Figure 3 ijms-22-00514-f003:**
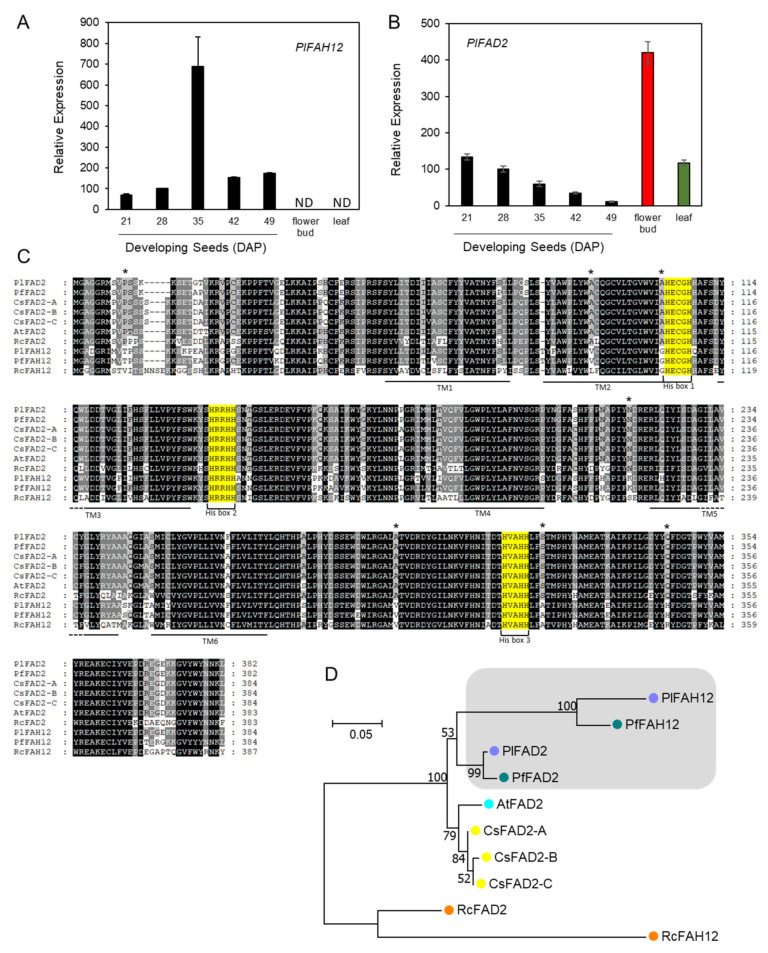
Characterization of PlFAH12s and PlFAD2s. Relative expression of *PlFAH12* (**A**) and *PlFAD2* (**B**) by qPCR. DAP, days after pollination. ND, not detected. (**C**) Amino acid alignment of FAH12 and FAD2 protein sequences. His-boxes were highlighted in yellow. Putative transmembrane domains (TM) were underlined. Amino acid residues identical in FAD2 but varied in FAH12 were marked with asterisks (*). (**D**) Phylogenetic tree of FAD2, FAH12 proteins from various species. Pl, *Physaria lindheimeri;* Pf, *Physaria fendleri*; At, *Arabidopsis thaliana*; Cs, *Camelina sativa*; Rc, *Ricinus communis*. Genbank accession numbers are listed in Appendix A.

**Figure 4 ijms-22-00514-f004:**
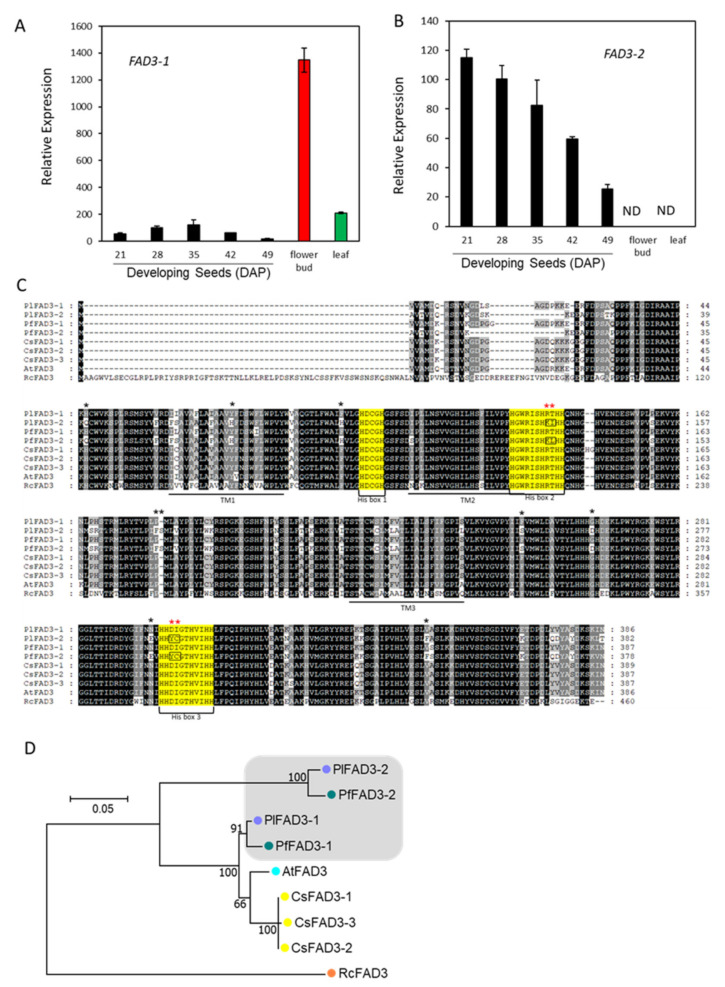
Characterization of FAD3s. Relative expression of *PlFAD3-1* (**A**) and *PlFAD3-2* (**B**) by qPCR. DAP, days after pollination. ND, not detected. (**C**) Amino acid alignment of FAD3 protein sequences. His-boxes were highlighted in yellow. Putative transmembrane domains (TM) were underlined. Asterisks indicate the aa that were conserved between PlFAD3-2 and PfFAD3-2, but that were different from the rest of the plant FAD3s. (**D**) Phylogenetic tree of FAD3 from various species. Pl, *Physaria lindheimeri*; Pf, *Physaria fendleri*; At, *Arabidopsis thaliana*; Cs, *Camelina sativa*; Rc, *Ricinus communis*. Genbank accession numbers were listed in Appendix A.

**Figure 5 ijms-22-00514-f005:**
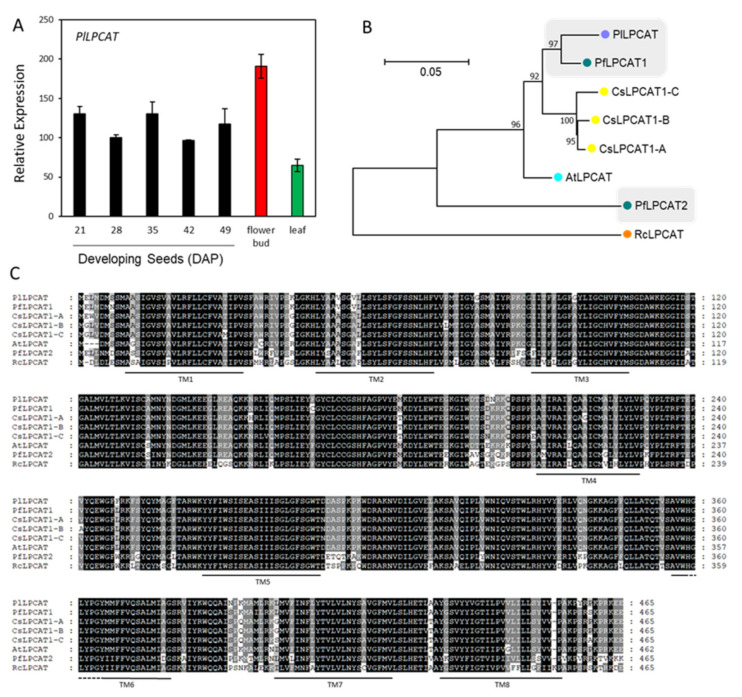
Characterization of LPCAT proteins. (**A**) Relative expression level of *PlLPCAT* in developing seed, flower, and leaf tissue by qRT-PCR. DAP; day after pollination. (**B**) Phylogenetic tree of LPCAT proteins in various species. *Physaria lindheimeri*, *Physaria fendleri* were highlighted with grey background. (**C**) Amino acid alignment of LPCAT protein sequence in *Physaria lindheimeri* (Pl), *Physaria fendleri* (Pf), *Arabidopsis thaliana* (At), *Camelina sativa* (Cs), and *Ricinus communis* (Rc). Putative transmembrane domains were underlined. Genbank accession numbers were listed in Appendix A.

**Figure 6 ijms-22-00514-f006:**
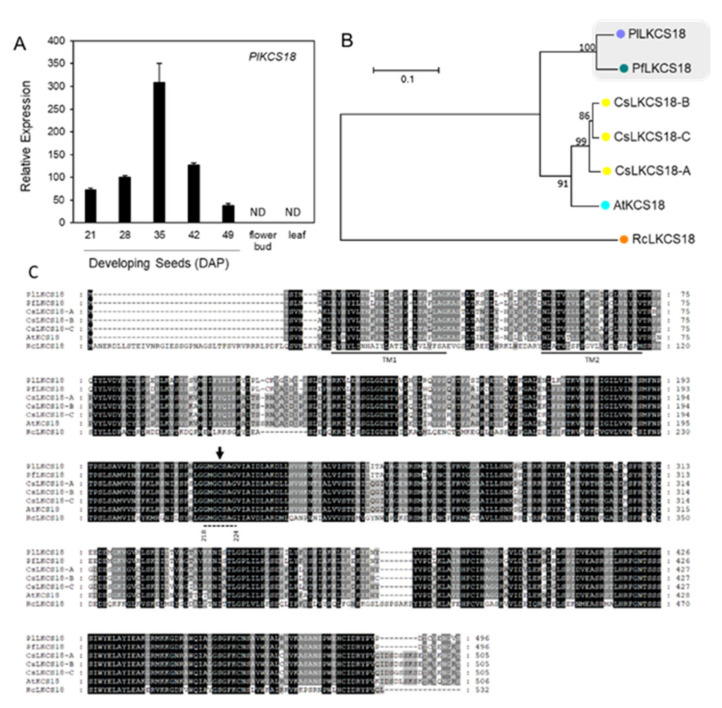
Characterization of KCS18 proteins. (**A**) Relative expression of *PlKCS18* by qRT-PCR. DAP, days after pollination. (**B**) Phylogenetic tree of KCS18 proteins from various species. Pl, *Physaria lindheimeri*; Pf, *Physaria fendleri*; At, *Arabidopsis thaliana*; Cs, *Camelina sativa*; Rc, *Ricinus communis*. (**C**) Amino acid alignment of KCS18 proteins. Putative transmembrane domains were underlined. Conserved region (dash-underlined between Gly218 and Gly224) containing an active Cys site among all KCS enzymes [71] was indicated by the arrow. Genbank accession numbers were listed in Appendix A.

**Figure 7 ijms-22-00514-f007:**
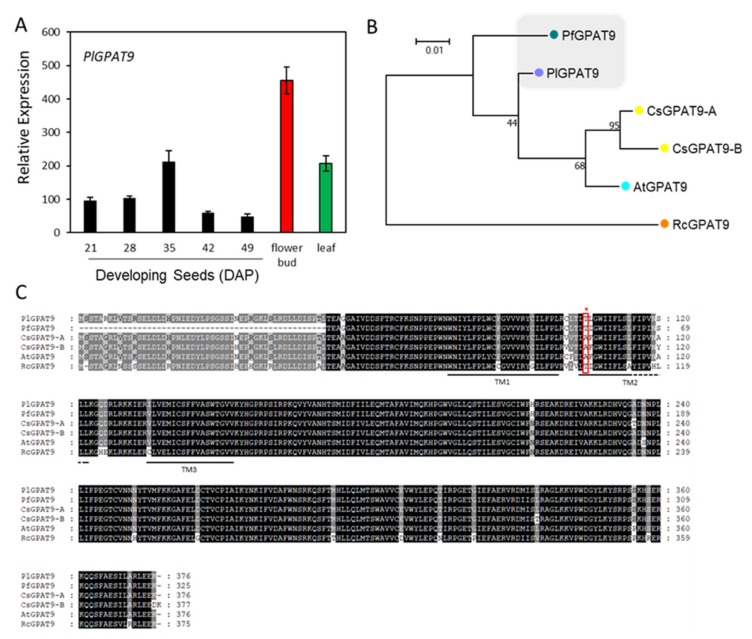
Characterization of GPAT9 proteins. (**A**) Relative expression level of *PlGPAT9* in developing seed, flower, and leaf tissues by qRT-PCR. DAP; day after pollination. (**B**) Phylogenetic tree of GPAT9 proteins in various species. Pl, *Physaria lindheimeri*; Pf, *Physaria fendleri*; At, *Arabidopsis thaliana*; Cs, *Camelina sativa*; Rc, *Ricinus communis*. PfDGAT9 and PlDGAT9 are highlighted in a grey background. (**C**) Amino acid alignment of GPAT9 protein sequences. Putative transmembrane domains are underlined. The red asterisk and box indicate the Thr^105^ of PlGPAT9 conserved among Pl, Pf, and Rc, but the corresponding position was Ala in At and Cs. Genbank accession numbers are listed in Appendix A.

**Figure 8 ijms-22-00514-f008:**
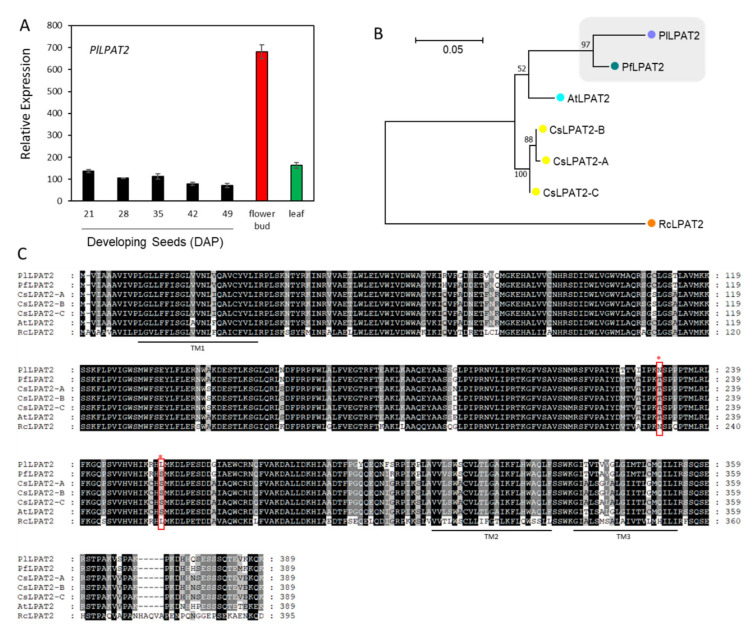
Characterization of LPAT2 proteins. (**A**) Relative expression level of *PlLPAT2* in the developing seed, flower, and leaf tissue by qRT-PCR. DAP; day after pollination. (**B**) Phylogenetic tree of LPAT2s in various species. Pl, *Physaria lindheimeri*; Pf, *Physaria fendleri*; At, *Arabidopsis thaliana*; Cs, *Camelina sativa*; Rc, *Ricinus communis*. PlLPAT2 and PfLPAT2 are highlighted with a grey background. (**C**) Amino acid alignment of LPAT2 protein sequences. Putative transmembrane domains are underlined. The 230th and 255th amino acids of PlLPAT2 are indicated by red asterisks and boxes. Genbank accession numbers are listed in Appendix A.

**Figure 9 ijms-22-00514-f009:**
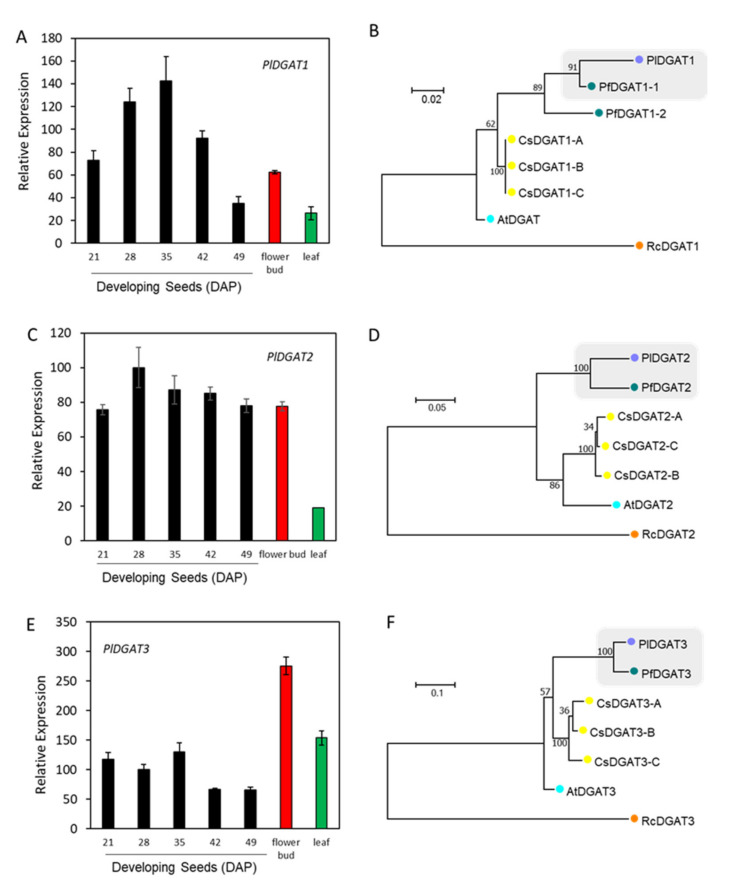
DGATs expression profiles and phylogenic trees. Relative expression level of *PlDGAT1* (**A**), *PlDGAT2* (**C**), and *PlDGA3* (**E**) in the developing seed, flower, and leaf tissue by qRT-PCR. DAP; day after pollination. Phylogenetic tree of proteins for DGAT1 (**B**), DGAT2 (**D**) and DGAT3 (**F**) from *Physaria lindheimeri* (Pl), *Physaria fendleri* (Pf), *Arabidopsis thaliana* (At), *Camelina sativa* (Cs) and *Ricinus communis* (Rc). Pl and Pf genes are highlighted with a grey background. Genbank accession numbers are listed in Appendix A.

**Figure 10 ijms-22-00514-f010:**
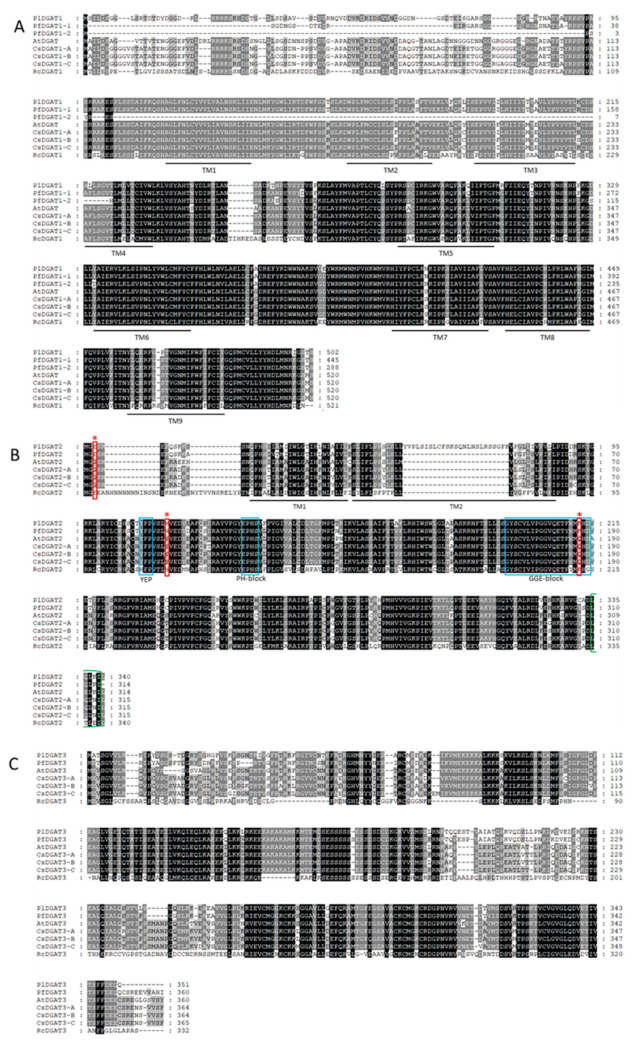
DGAT protein sequences alignment. Amino acid alignment of DGAT1 (**A**), DGAT2 (**B**), and DGAT3 (**C**). Putative transmembrane domains are underlined. The red asterisks and boxes indicate the amino acid corresponding to the 3rd, 115th, and 212th of PlDGAT2. The conserved YFP motif [108], PH-block and GGE-block [98] in plant DGAT2s are indicated in blue boxes. The modestly conserved ER retrieval motif in plant DGAT2s is in green brackets [103]. Abbreviation: Pl, *Physaria lindheimeri*; Pf, *Physaria fendleri*; At, *Arabidopsis thaliana*; Cs, *Camelina sativa*; Rc, *Ricinus communis*. Genbank accession numbers were listed in Appendix A.

**Figure 11 ijms-22-00514-f011:**
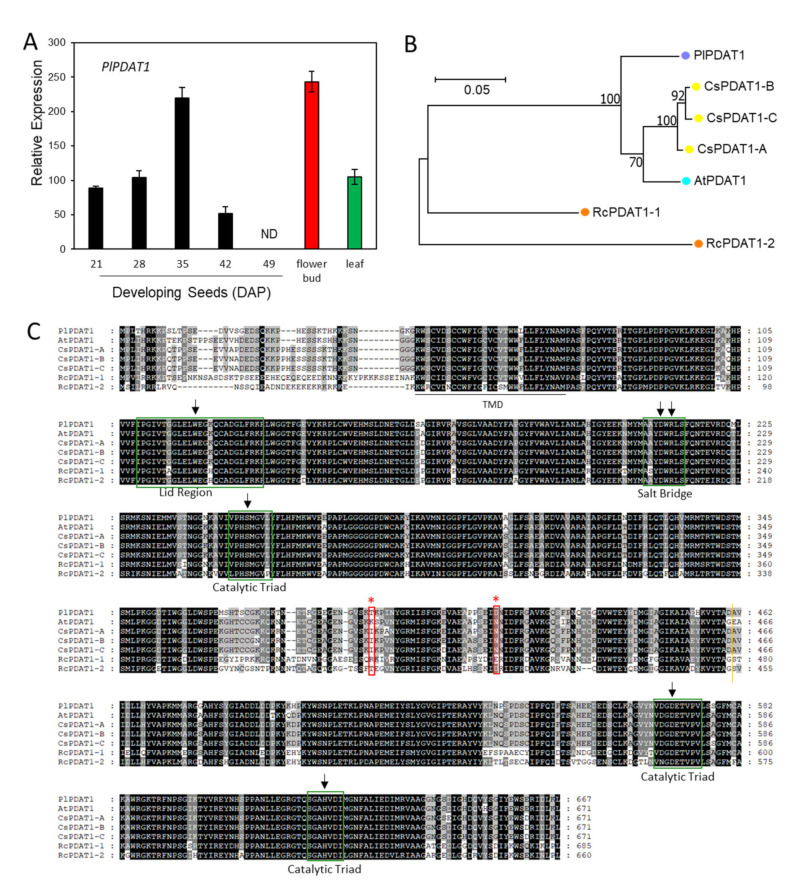
Characterization of PDAT1s in various species. (**A**) Relative expression level of *PlPDAT1* in the developing seed, flower, and leaf tissue by qRT-PCR. DAP; day after pollination. (**B**) Phylogenetic tree of PDAT1 proteins from *Physaria lindheimeri* (Pl), *Arabidopsis thaliana* (At), *Camelina sativa* (Cs), *Ricinus communis* (Rc). (**C**) Protein alignment of PDAT1s. Putative transmembrane domains are underlined. The Thr and Asp corresponding to the aa at the 390th and 415th positions of PlPDAT1 are indicated by red asterisks and boxes. Green boxes indicate lid region, salt bridge, and catalytic triad [126], where conserved Trp (W), Asp (D), Arg (R), and three coordinated amino acids Ser-Asp-His (or S-D-H) are indicated by arrows. Genbank accession numbers are listed in Appendix A.

**Figure 12 ijms-22-00514-f012:**
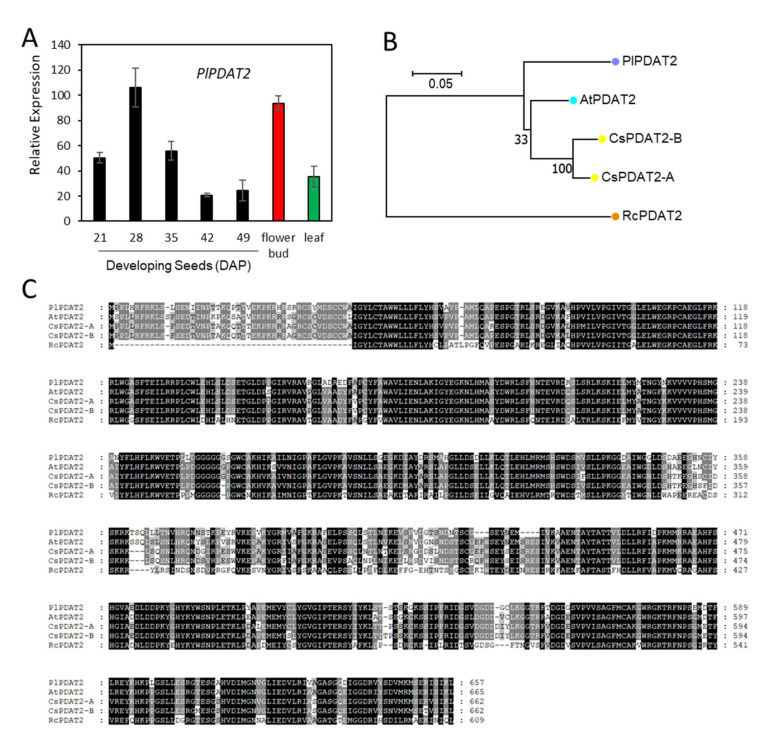
Characterization of PDAT2s in various species. (**A**) Relative expression level of *PlPDAT2* in the developing seed, flower, and leaf tissue by qRT-PCR. DAP; day after pollination. (**B**) Phylogenetic tree of PDAT2 proteins from *Physaria lindheimeri* (PI), *Arabidopsis thaliana* (At), *Camelina sativa* (Cs), *Ricinus communis* (Rc). (**C**) Protein sequences alignment of PDAT2s. Genbank accession numbers are listed in Appendix A.

**Figure 13 ijms-22-00514-f013:**
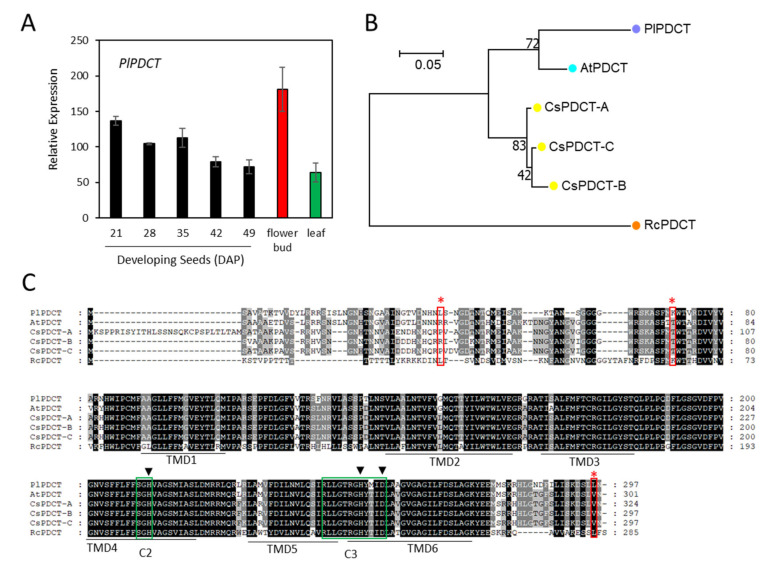
Characterization of PDCTs in various species. (**A**) Relative expression level of *PlPDCT* in the developing seed, flower, and leaf tissue by qRT-PCR. DAP; day after pollination. (**B**) Phylogenetic tree of PDCT proteins from *Physaria lindheimeri*, (Pl), *Arabidopsis thaliana* (At), *Camelina sativa* (Cs), *Ricinus communis* (Rc). (**C**) Protein alignment of PDCTs. Putative transmembrane domains were underlined. The red asterisks and boxes indicate the amino acids corresponding to the Leu, Lys, and Leu at the 40th, 71th and 295th positions of PlPDCT. Green boxes are the C2 and C3 domains, where a set of three coordinated amino acids, H-H-D (or His-His-Asp), of a catalytic triad [128] are indicated by arrows. Genbank accession numbers are listed in Appendix A.

**Table 1 ijms-22-00514-t001:** Summary of sequencing data of *P. lindheimeri* seeds, flower bud and leaf transcriptomes.

	Developing Seed (DAP) ^a^			Flower Bud	Leaf	Total
	21	35	42			
raw reads	129,121,576	99,873,058	127,855,980	99,729,366	96,034,602	552,614,582
clean reads ^b^	125,026,944	97,400,124	123,850,518	98,065,392	95,366,514	539,709,492
ratio ^c^ (%)	96.8	97.5	96.9	98.3	99.3	97.7

^a^ days after pollination. ^b^ clean reads were generated by trimming the adapter and low-quality reads. ^c^ ratio is calculated by clean reads/raw reads.

**Table 2 ijms-22-00514-t002:** Statistical summary of de novo transcriptome assembly.

	Number
Numbers of unique genes transcripts (longest isoforms)	85,948
Numbers of transcripts (contigs)	129,633
Average length of transcripts (bp)	814 (bp)
N50 ^a^ transcript size (bp)	1373 (bp)
Total assembled bases	105,504,114 (nt)

^a^ the N50 is defined as the minimum contig length needed to cover 50% of the genome.

## Data Availability

Sequence raw data of Transcriptomes was registered in the NCBI Sequence Read Archive (SRA) (http://www.ncbi.nlm.nih.gov/), under the BioProject: PRJNA672105. All data was shared in Appendix A.

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
