# Peer review of "Transcriptome Analysis and Identification of Lipid Genes in Physaria lindheimeri, a Genetic Resource for Hydroxy Fatty Acids in Seed Oil"

_ijms, 2021, doi:10.3390/ijms22020514_

Round 1

Reviewer 1 Report

The manuscript of Chen et al describe their efforts to discover lipid genes involved in HFA synthesis in Physaria lindheimeri. To this end, transcripts from developing seeds at various stages, as well as leaf and flower bud were sequenced. Characterization of key genes by quantitative PCR (qPCR) is described. They identified 40 P. lindheimeri genes involved in fatty acid and seed oil biosynthesis, 37 of them shared 78–100% nucleotide identity with Arabidopsis orthologs. Two genes, PlFAH12 and PlKCS18, directly responsible for 20:1OH synthesis, were expressed only in developing seeds and showed high peaks at 35 DAP, coinciding with the maximum 20:1OH accumulation rate during seed development. The information described in the present manuscript may provide a valuable resource for the biotechnological production of HFA in existing oilseed crops.

Given the industrial applications of HFAs, these results will be of interest for the readers, and thus, this manuscript merits publication.

I recommend minor revision before publication in International Journal of Molecular Sciences.

Comments

  1. Abstract. “Hydroxy fatty acid (HFA) has numerous industrial applications but is absent in most vegetable oils.” I suppose they mean plural, hydroxy fatty acids (HFAs). Please clarify throughout the text if you talk about one particular HFA, or a family of compounds.
  2. Two very recent publications report biological activities for HFAs. Hydroxystearic acids inhibit the cell growth of human cancer cell and suppress the beta-cell apoptosis (J. Med. Chem. 2020, 63, 12666–12681). In addition, families of HFAs have been recognized as minor components in milk (Molecules 2020, 25, 3947). The authors can discuss these data in the introduction highlighting the importance of HFAs.
  3. Could the approach described in the present manuscript direct the research for the production of bioactive HFAs in plants? The authors could discuss it in the discussion section.

Author Response

Our response to #1 reviewer:

  1. Abstract. “Hydroxy fatty acid (HFA) has” is replaced with “Hydroxy fatty acids (HFAs) have” (line 18). We also have checked and clarified the concept ‘HFA’ or ‘HFAs’ throughout the manuscript.
  2. The two publications and related info suggested by Reviewer#1 are cited in revision line 51–55.

3. We include the related info in Conclusion (lines 869).

Reviewer 2 Report

The manuscript by Chen et al. reports the transcriptome profiling of Physaria lindheimeri that accumulates high amounts of hydroxy fatty acid. The authors identified and extensively discussed lipid genes that may be crucial for seed hydroxy fatty acid accumulation from the transcriptome data. The study provides important information on hydroxy fatty acid accumulation in P. lindheimeri and possible candidate genes for metabolic engineering of hydroxy fatty acid production. The manuscript is well written and reads smoothly. I only have a few suggested improvements.

Major comments:

  1. I appreciate the extensive comparison of crucial TAG biosynthetic enzymes from P. lindheimeri with orthologs from species that do or do not accumulate hydroxy fatty acids. It is very useful and interesting as Arabidopsis and Physaria are evolutionarily closely related and some Arabidopsis and Physaria enzymes share very high sequence identity but likely have different substrate specificity. The sequence alignment results are interesting, and it might provide more information (although it would be speculative) if the authors may further compare the differences in the predicted structure models (Phyre2 or SWISS Model) of a couple of selected enzymes (eg. FAD2/FADH or PDAT) if possible and make some comments in the discussion.
  2. The transcriptome data are informative, but RNA levels do not always directly represent the metabolic fluxes and enzyme activities, especially considering the presence of post-transcriptional/translational regulation. Therefore, it would be worthwhile if the authors could comment on this in the discussion.

Minor comments and typos:

  1. Throughout the text: use italic type for genes and transcripts, and normal type for enzymes. Eg. lines 556-559.
  2. Throughout the text: check the use of abbreviations. Eg. fatty acid and FA, triacylglycerol and TAG
  3. Line 59: “forward” reaction… resulting “in”
  4. Line 167: resulted “in”
  5. Line 243: De novo “fatty acids are”
  6. Line 385: “Reverse” reaction
  7. Lines 444 and 446: “PlLACS”
  8. Line 596: “and”
  9. Line 633: “18:1OH”
  10. Lines 636, 651 and 652: Please check the nomenclature of BnDGAT2 isoforms.
  11. Line 647: “whose” function
  12. Line 670: “polyunsaturated”
  13. Line 672: Please give the scientific name of sesame.
  14. Line 676: “patterns”
  15. Please double check the references. Eg. ref 50 and 74 are the same.

Author Response

Jan. 1st, 2021

Dear Editor-in-Chief and Reviewer2,

Thank you very much for taking care of our manuscript (ijms-1021173). I would also like to thank Reviewers for their time and comments. Our itemized answers are next to the reviewers’ comments below in Bold fonts.

Besides, we added a recent discovery of AtPLDζs gene family to update our manuscript (revision line 756–762) and the following revisions are made to accommodate the updates:

1) Abstract, line 25–26, “40” and “37” are replaced with “42” and “39”, respectively.

2)A new Sup file 13, Figure S13 is added to support the PlPLDζs data; the original Sup file 13, Table S19 is changed to current Sup file 14, Table S19;

3) We revised Sup file 2, Table S1 to S5 and Sup file 3, Table S6 to include the updated PlPLDζs contigs info.

4) To avoid errors, we have resubmitted all Sup files as one package (revision Sup files zip).

All revised texts are in red fonts in revision ijms-1021173.

We have submitted revision ijms-1021173, “Transcriptome Analysis and Identification of Lipid Genes in Physaria lindheimeri, a Genetic Resource for Hydroxy Fatty Acids in Seed Oil”, for publication in IJMS Special issue, "Biological Networks of Specialized Metabolites and Plants".

Best regards

Grace Chen, Ph. D.

Research Plant Physiologist

USDA, ARS, WRRC

Bioproducts Research Unit

800 Buchanan St

Albany, CA 94710

Cell 510-417-1979; Office: 510-559-5627

Fax: 510-559-5768

-----------------------------------------------------------------------------------------------------------

#2 reviewer

The manuscript by Chen et al. reports the transcriptome profiling of Physaria lindheimeri that accumulates high amounts of hydroxy fatty acid. The authors identified and extensively discussed lipid genes that may be crucial for seed hydroxy fatty acid accumulation from the transcriptome data. The study provides important information on hydroxy fatty acid accumulation in P. lindheimeri and possible candidate genes for metabolic engineering of hydroxy fatty acid production. The manuscript is well written and reads smoothly. I only have a few suggested improvements.

Major comments:

  1. I appreciate the extensive comparison of crucial TAG biosynthetic enzymes from P. lindheimeri with orthologs from species that do or do not accumulate hydroxy fatty acids. It is very useful and interesting as Arabidopsis and Physaria are evolutionarily closely related and some Arabidopsis and Physaria enzymes share very high sequence identity but likely have different substrate specificity. The sequence alignment results are interesting, and it might provide more information (although it would be speculative) if the authors may further compare the differences in the predicted structure models (Phyre2 or SWISS Model) of a couple of selected enzymes (eg. FAD2/FADH or PDAT) if possible and make some comments in the discussion.

  1. The transcriptome data are informative, but RNA levels do not always directly represent the metabolic fluxes and enzyme activities, especially considering the presence of post-transcriptional/translational regulation. Therefore, it would be worthwhile if the authors could comment on this in the discussion.

Minor comments and typos:

  1. Throughout the text: use italic type for genes and transcripts, and normal type for enzymes. Eg. lines 556-559.
  2. Throughout the text: check the use of abbreviations. Eg. fatty acid and FA, triacylglycerol and TAG
  3. Line 59: “forward” reaction… resulting “in”
  4. Line 167: resulted “in”
  5. Line 243: De novo “fatty acids are”
  6. Line 385: “Reverse” reaction
  7. Lines 444 and 446: “PlLACS”
  8. Line 596: “and”
  9. Line 633: “18:1OH”
  10. Lines 636, 651 and 652: Please check the nomenclature of BnDGAT2 isoforms.
  11. Line 647: “whose” function
  12. Line 670: “polyunsaturated”
  13. Line 672: Please give the scientific name of sesame.
  14. Line 676: “patterns”
  15. Please double check the references. Eg. ref 50 and 74 are the same.

Our response to #2 reviewer:

  1. Based on Reviewer#2’s suggestion, we have analyzed the structure models of AtFAD2, PIFAD2 and PIFAH12 using the web tools, SWISS, https://swissmodel.expasy.org/interactive and Phyre2, http://www.sbg.bio.ic.ac.uk/~phyre2/html/page.cgi?id=index. We did not find any differences among these three proteins’ structures. The main reason is probably that both Phyre2 and SWISS model are homology based modeling and use already existed information from Protein Data Base as template. When we used AtFAD2, PIFAD2 and PIFAH12 as queries, our query proteins were, however, compared with a human soluble acyl-CoA desaturase. As our queries are membrane bound enzymes, we don’t think Phyre2 and SWISS models predict precisely for the structure AtFAD2, PIFAD2 and PIFAH12. Since AtFAD2, PIFAD2 and PIFAH12 share high sequence homology, there could be limits in discriminating these three proteins by Phyre2 and SWISS 3D models. Considering these factors, we would not include the results in Discussion.

We appreciate Reviewer #2’s comments, and will follow advancement of technology in future to analyze the substrate specificity of enzymes using 3D models.

  1. Based on the #2 reviewer’s comments, we include related info, genome-wide proteotypes analysis and metabolic flux analysis, in revision line 863–866.

  1. Minor comments and typos:

  1. 1. We have checked and corrected accordingly throughout the manuscript to use italic type for genes and transcripts, and normal type for enzymes.

  1. We have checked and corrected accordingly throughout the manuscript to use correct abbreviations, ie., FA for fatty acid and TAG for triacylglycerol.

  1. “forwarding” is replaced with “forward” (revision line 62); “in” is added before “resulting” (revision line 63)

  1. “in” is added after “resulted” (revision line 170)

  1. Correct words “De novo FAs” are used in revision line 245.

  1. “Backward” is replace with “Reverse” in revision line 387.

  1. “S” is added to “PlLACS” revision line 446–448.

  1. “ant” is corrected to “and” in revision line 599.

  1. corrected to “epoxy vernolic FA (18:1>O)” in revision line 636.

  1. Revision line 638–639, the nomenclature of BnDGAT2 isoforms are correct based on the publication [103].

  1. “which” is replaced with “whose” in revision line 650.

  1. “poly unsaturated” is corrected to “polyunsaturated” in revision line 673.

  1. The scientific name of sesame, Sesamum indicum, is added in revision line 675.

  1. “patterns” is corrected in revision line 680.

  1. the correct ref [52] is used in revision lines 127, 433, 444, 447 and 451.